# Cyclophilin J limits inflammation through the blockage of ubiquitin chain sensing

Chunjie Sheng [1], Chen Yao[1], Ziyang Wang[1], Hongyuan Chen[2], Yu Zhao[3], Dazhi Xu[1], Haojie Huang[3], Wenlin Huang[1] & Shuai Chen[1]

Maintaining innate immune homeostasis is important for individual health. Npl4 zinc finger (NZF) domain-mediated ubiquitin chain sensing is reported to function in the nuclear factor-kappa B (NF-κB) signal pathway, but the regulatory mechanism remains elusive. Here we show that cyclophilin J (CYPJ), a member of the peptidylprolyl isomerase family, is induced by inflammation. CYPJ interacts with the NZF domain of transform growth factor-β activated kinase 1 binding protein 2 and 3 as well as components of the linear ubiquitin chain assembly complex to block the binding of ubiquitin-chain and negatively regulates NF-κB signaling. Mice with *Cypj* deficiency are susceptible to lipopolysaccharide and heat-killed *Listeria monocytogenes*-induced sepsis and dextran sulfate sodium-induced colitis. These findings identify CYPJ as a negative feedback regulator of the NF-κB signaling pathway, and provide insights for understanding the homeostasis of innate immunity.

[1] Sun Yat-sen University Cancer Center, State Key Laboratory of Oncology in South China, Collaborative Innovation Center of Cancer Medicine, 510060 Guangzhou, Guangdong, China. [2] Department of Pathogen Biology and Immunology, School of Basic Course, Guangdong Pharmaceutical University, 510006 Guangzhou, Guangdong, China. [3] Department of Biochemistry and Molecular Biology, Mayo Clinic College of Medicine, Rochester, MN 55905, USA. These authors jointly supervised this work: Shuai Chen and Wenlin Huang. Correspondence and requests for materials should be addressed to S.C. (email: shuaichen2010@hotmail.com)

Pathogen infections are recognized by a series of pattern recognition receptors (PRRs) that trigger the expression of a series of cytokines for host defense[1]. Proper regulation of the innate immune response is critical to the host[1]. As a versatile post-translational modification, ubiquitination is commonly used as a regulatory mechanism to control diverse processes of the immune system, including innate immune recognition and the clearance of pathogens[2–4]. A protein can be modified on one lysine residue with a single ubiquitin (monoubiquitylation) or with a chain of ubiquitins (polyubiquitylation)[5]. Lysine 48 (K48)-linked polyubiquitylation usually targets proteins for proteasomal degradation, whereas K63-linked polyubiquitylation is important in regulating signal transduction cascades[6]. The function of polyubiquitin chains linked through other lysines of ubiquitin remains largely unknown. K63-linked polyubiquitin chains are assembled in the cytosol after the stimulation of cells with cytokines or Toll-like receptor (TLR) ligands[7]. K63-linked chains recruit and activate a protein kinase complex containing transforming growth factor β-activated kinase 1 (TAK1), TAK1-binding protein 1 (TAB1), and TAB2 or its homolog TAB3[8,9]. This TAK1 complex has a critical function in the activation of the nuclear factor-kappa B (NF-κB) and Jun N-terminal kinase (JNK) signal pathways[10]. TAK1 directly phosphorylates both IκB kinase-β (IKK-β) and MKK6 to stimulate the NF-κB and JNK pathways, respectively[11]. The activated IKK complex (including IKK-α, IKK-β, and NEMO) phosphorylates IκBα, which undergoes K48-linked polyubiquitination and subsequent degradation, releases NF-κB transcription factor heterodimers (such as p50 and p65) from the cytoplasm to the nucleus and initiates the transcription of proinflammatory cytokines[12,13].

The TAK1-binding proteins (TAB1, TAB2, and TAB3) play important roles in the activation of downstream signals. TAB1 interacts with the N-terminus of of TAK1 and acts as a coactivator[14,15]. TAB2 and TAB3 have redundant functions in vivo and act as adapters to recruit TAK1 and TAB1 to K63-linked polyubiquitin chains for activation. The C-terminal Npl4 zinc-finger (NZF) domain of TAB2 and TAB3 specifically recognizes K63-linked polyubiquitin chains that are unanchored or anchored to the substrate proteins such as RIP1[16–19]. This recognition is essential for the TAK1-induced activation of the NF-κB and JNK pathways. NZF domains are typically ~30-residue protein-protein interaction modules that coordinate a single zinc ion with four conserved cysteine residues[20]. Two cysteines, C670 and C673, are important for the interaction of TAB2 with K63-Ub, whereas only C706 is critical for TAB3[21]. Most of the ubiquitin-binding NZF domains bind polyubiquitin chains without apparent specificity, whereas the NZF domains of TAB2 and TAB3 specifically bind K63-linked polyubiquitin chains[21–24]. NZF domains have also been identified in members of the recently identified linear ubiquitin-chain assembly complex (LUBAC), such as HOIP, HOIL-1L, and SHARPIN, and they are critical for the assembly of the complex and specific binding to linear ubiquitin chains[25]. Disturbing the interaction between NZF domains and polyubiquitin chains plays an important role in the regulation of innate immune responses. For example, modification of TAB2 and TAB3 NZF domains by the bacterial methyltransferase NleE[26] or occupation of the TAB3 NZF by mycobacterium PtpA[27] disrupts the recruitment of K63-Ub chain and thus attenuates NF-κB signaling and host defense.

Cyclophilins (CYPs) belong to the peptidyl-prolyl cis-trans isomerase (PPIase) chaperon family[28]. CYPs were initially identified as intracellular receptors of immunosuppressive drug cyclosporine A (CsA), and they participate in diverse biological functions such as protein folding, signal transduction, apoptosis, inflammation, tumorigenesis, and viral infection[28,29]. As a novel member of CYPs, human cyclophilin J (CYPJ; also named

PPIase-like 3, PPIL3) shows approximately 50% sequence similarity to human CYPA and has an unknown function[30–32].

Here, we explore the function of CYPJ in innate immune regulation. We show that CYPJ protein is induced in response to inflammatory stimulations, and is recruited to the NZF domains of TAB2 and TAB3 via its C-terminal region. CYPJ disrupts the binding of the K63-Ub chain to TAB2 and TAB3, attenuates the NF-κB signal pathway, and rescues mice from lethal endotoxin shock and dextran sulfate sodium (DSS)-induced colitis. These findings identify CYPJ as an unexpected negative feedback regulator of the NF-κB signaling pathway and a potential target for developing anti-inflammation agents.

## Results

**Inflammation-induced CYPJ inhibits the NF-κB signaling.** To identify new regulators of innate immune responses, we performed a luciferase based gene screen in 293T cells[33], and the results showed that CYPJ is a negative regulator of tumor necrosis factor (TNF) induced activation of the NF-κB reporter (Fig. 1a). As NF-κB can be triggered by diverse innate immune pathways, we monitored their regulation by CYPJ, and the results showed that enforced CYPJ expression greatly repressed NF-κB activation induced by MAVS and MyD88 but not TRIF (Supplementary Fig. 1a). In addition to the NF-κB reporter, CYPJ overexpression also suppressed the transcription of NF-κB targets genes *TNF* and *IL-8* induced by TNF (Fig. 1b) or A/WSN/1933 influenza A virus infection (termed WSN for short; Supplementary Fig. 1b). Interestingly, CYPA and catalytically dead mutants of both CYPJ and CYPA[34] repressed TNF-induced NF-κB activation to the level similar to that of CYPJ (Supplementary Fig. 1c), demonstrating that the inhibitory function of these cyclophilins is independent of their PPIase enzymatic activity. The knockout of *CYPJ* by CRISPR-Cas9-mediated genomic edition in 293T cells (Fig. 1c, inset) significantly increased TNF-induced activation of the NF-κB reporter (Fig. 1c), as well as the transcription of *TNF* and *IL-8* (Fig. 1d). We also silenced CYPJ expression by two independent short interference RNA (siRNA; Supplementary Fig. 1d), which significantly increased TNF-induced activation of the NF-κB reporter (Supplementary Fig. 1e). For comparison, we also knocked out CYPA by CRISPR-Cas9 in 293T cells (Supplementary Fig. 1f, inset). CYPA deficiency enhanced both TNF-induced NF-κB reporter activation (Supplementary Fig. 1f) and transcription of *TNF* and *IL-8* (Supplementary Fig. 1g).

As classic NF-κB activation largely dependents on the phosphorylation of the IKK complex which induces the degradation of IκBα and the nuclear translocation of NF-κB transcription factors[7], we further elucidated the function of CYPJ in this process. We transfected 293T cells with or without HA-CYPJ and treated the cells with TNF (Fig. 1e) or infected the cells with WSN virus (Supplementary Fig. 1h). As predicted, CYPJ attenuated the TNF-induced phosphorylation of IKKα/β and IκBα (Fig. 1e), as well as the WSN infection-induced phosphorylation of IKKα/β (Supplementary Fig. 1h). In another assay, we treated *CYPJ* knockout (Fig. 1f) or *CYPA* knockout (Supplementary Fig. 1i) 293T cells with TNF, and the results showed that deficiency of CYPJ or CYPA significantly increased TNF-induced phosphorylation of IKKα/β and IκBα (Fig. 1f, Supplementary Fig. 1i). Immunofluorescence analyses revealed that enforced expression of GFP-fused CYPJ (GFP-CYPJ) inhibited the TNF-stimulated translocation of p65 from the cytoplasm to the nucleus (Fig. 1g), which was confirmed in nuclear/cytoplasm fractionation experiments (Fig. 1h).

Considering that innate immune responses are tightly regulated via negative feedback[4], we evaluated the expression of CYPJ and CYPA upon inflammatory stimulations. Western blots

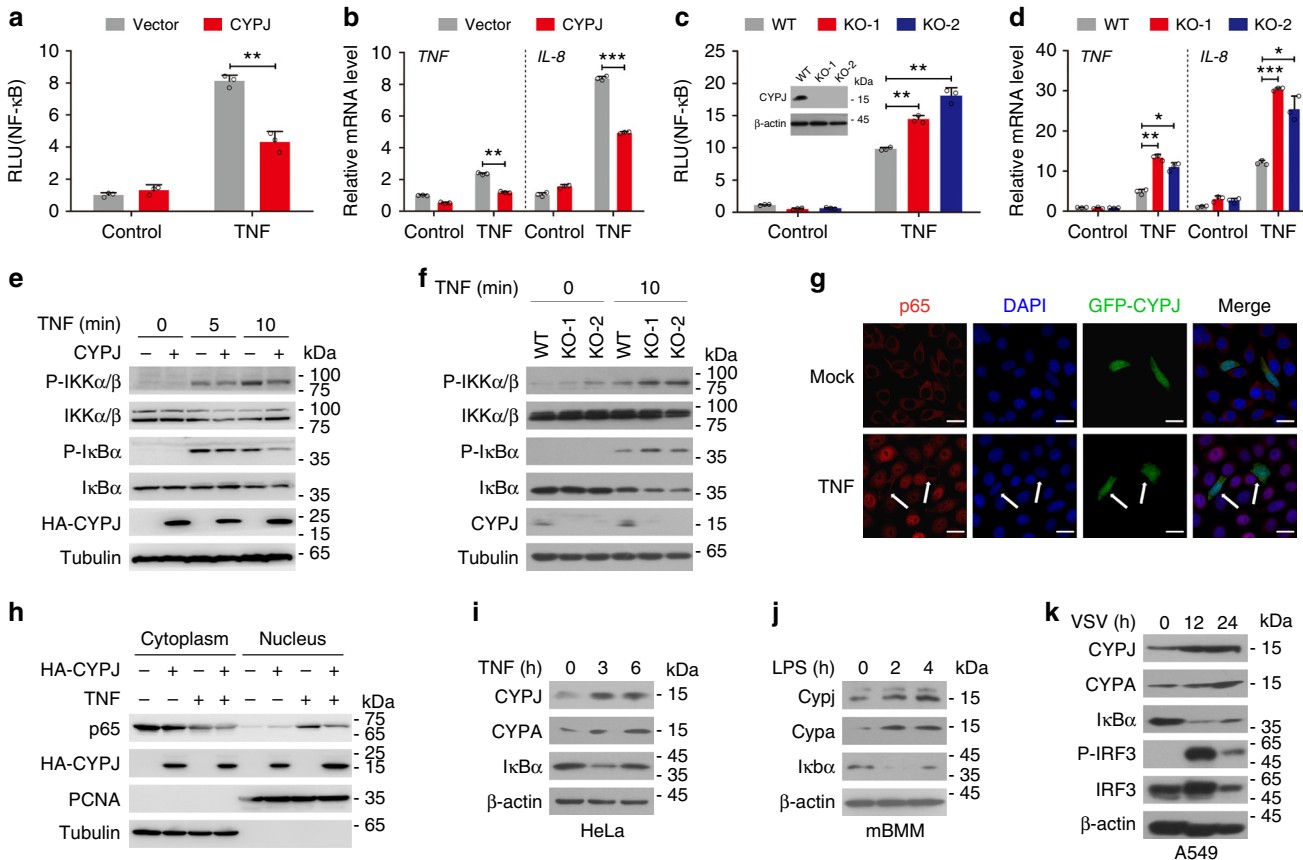

**Fig. 1** CYPJ negatively regulates the NF-κB signal pathway. **a** CYPJ inhibits TNF-induced activation of the NF-κB reporter in 293T cells (N = 3). **b** CYPJ inhibits TNF-induced transcription of *TNF* and *IL-8* in 293T cells (N = 3). **c** CRISPR-Cas9-mediated CYPJ deficiency (inset) enhances TNF-induced activation of the NF-κB reporter in 293T cells (N = 3). **d** CYPJ knockout enhances TNF-induced transcription of *TNF* and *IL-8* in 293T cells (N = 3). **e** 293T cells were transfected with a control or HA-CYPJ plasmid and were treated with TNF for different times. Phosphorylation of the indicated proteins was detected. **f** WT and CYPJ-deficient 293T cells were treated with or without TNF, and phosphorylation of the indicated proteins was detected. **g** HeLa cells were transfected with pEGFP-CYPJ (green) and treated with or without TNF (15 ng ml$^{-1}$) for 15 min. The cells were fixed, permeabilized, and stained for p65 (red) and 4′6-diamidino-2-phenylindole (DAPI) (blue), followed by observation by laser confocal fluorescence microscopy. Scale bars, 20 μm. The white arrows indicate GFP-CYPJ transfected cells. **h** 293T cells were transfected with or without HA-CYPJ and stimulated by TNF (15 ng ml$^{-1}$) for 15 min. The cytoplasmic and nuclear proteins were isolated and detected with the indicated antibodies. **i–k** Protein levels of CYPJ and CYPA were detected in TNF-treated HeLa cells (**i**), LPS-treated primary mBMM cells (**j**), and VSV-infected A549 cells (**k**). Error bars indicate the standard deviation (S.D.); n.s. no significance, * $p < 0.05$, ** $p < 0.01$, *** $p < 0.001$ (two-tailed Student's $t$-test)

showed that the protein levels of both CYPJ and CYPA were elevated in TNF-treated HeLa cells (Fig. 1i), lipopolysaccharide (LPS)-treated primary murine bone marrow derived macrophages (mBMM, Fig. 1j) and vesicular stomatitis virus (VSV)-infected A549 cells (Fig. 1k). Interestingly, the mRNA abundance of *CYPJ* and *CYPA* remained unchanged under all conditions (Supplementary Fig. 2a-c). As internal controls, the transcription of *TNF* and *IL-8* or *Il-6* were significantly induced (Supplementary Fig. 2a-c). These results demonstrated that CYPJ and CYPA are induced at the post-transcriptional level in response to inflammation stimuli and negatively regulates NF-κB activation.

**CYPJ attenuates the NF-κB signaling in primary mouse BMMs.** Subsequently, we generated *Cypj*-knockout mice to assess the function of this gene in primary cells. The third exon of mouse *Cypj* is edited by CRISPR-Cas9 to generate an in-frame stop codon at the 18$^{th}$ amino acids (Fig. 2a), which results in complete depletion of the Cypj protein in mBMMs (Supplementary Fig. 3a). *Cypj* deficiency did not affect the percentage of mBMMs (Fig. 2b) or mouse bone marrow derived dendritic cells

(mBMDC, Fig. 2c). Further analyses did not identify developmental defects of major innate and adaptive immune cells in Cypj-deficient mice (Supplementary Fig. 3b). We treated wild-type and *Cypj*-knockout mBMMs with LPS, Tnf, heat-killed *Listeria monocytogenes* (HKLM), and Il-1β for different times and detected the phosphorylation of Ikkα/β, Ikbα, Jnk, and p38 by western blots and the transcription of *Tnf* and *Il-6* by qRT-PCR. In accordance with the findings from cell lines, Cypj deficiency increased the phosphorylation of these kinases in NF-κB and JNK pathways and the mRNA abundance of *Tnf* and *Il-6* in response to LPS (Fig. 2d, e), Tnf (Fig. 2f, g), HKLM (Fig. 2h, i), and Il-1β (Supplementary Fig. 3c,d). In another assay, we knocked down Cypj and Cypa by siRNA in murine BMMs (Supplementary Fig. 4a,b), which resulted in augmented secretion of Tnf and Il-6 (Supplementary Fig. 4c,d). Unexpectedly, Cypj deficiency did not affect LPS- or VSV-induced IRF3 phosphorylation or *IFN-β* transcription in mBMMs (Supplementary Fig. 5a-c), suggesting that Cypj does not alter the signal pathway that controls type I interferon transcription. The above results confirm that Cypj, as well as Cypa attenuates NF-κB signaling and proinflammatory cytokine expression in murine primary cells.

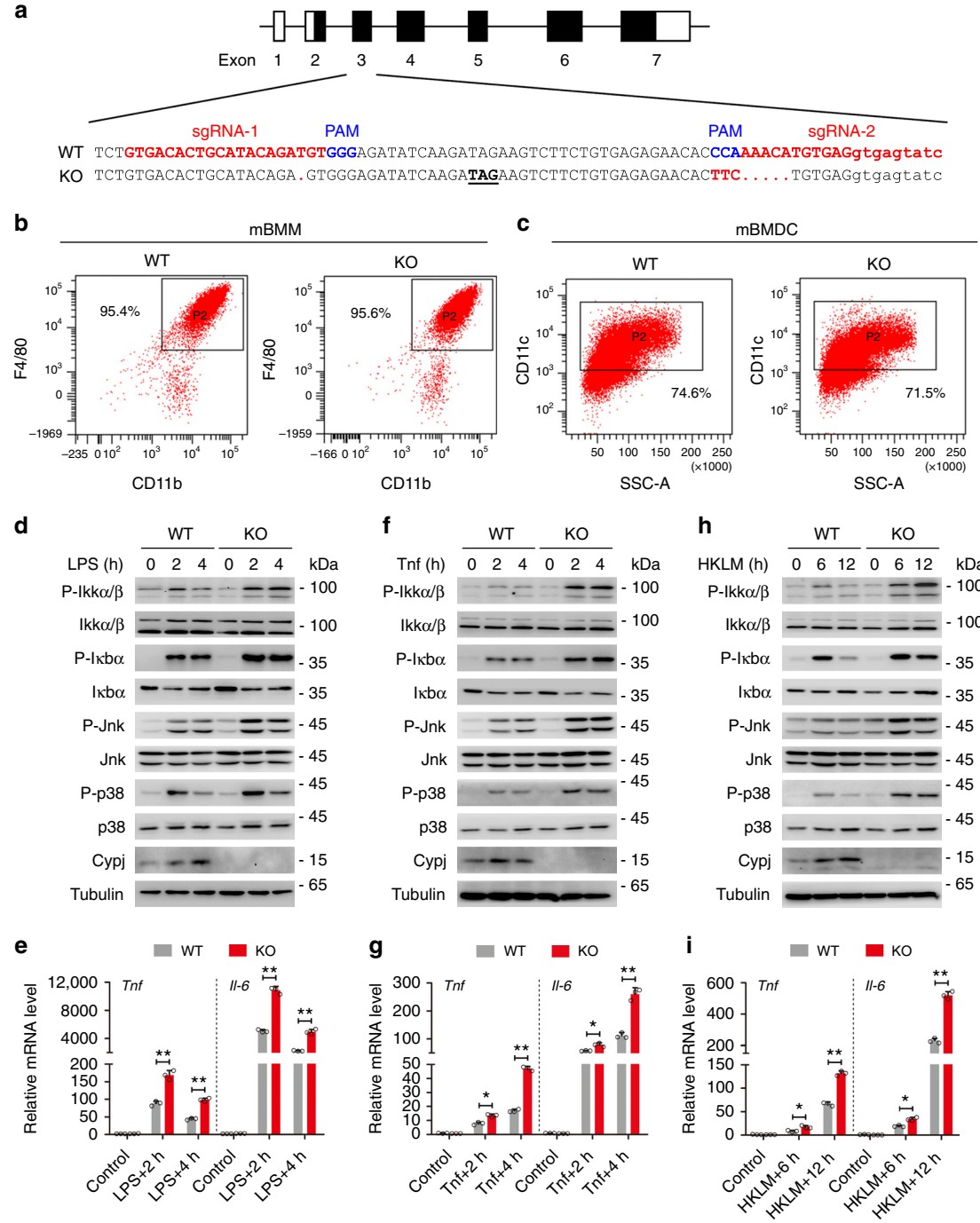

**Fig. 2** Cypj deficiency activates the NF-κB signaling in primary mBMM cells. **a** The strategy of CRISPR-Cas9 mediated genomic edition. A gene specific guide RNA (sgRNA) was used to achieve the targeting. **b**, **c** Percentage of mBMMs (**b**) and mBMDC (**c**) were analyzed by their specific lineage markers between WT and KO mice. **d–i** WT and Cypj-deficient primary mouse BMM cells were treated with or without LPS (100 ng ml$^{-1}$) (**d**, **e**), Tnf (20 ng ml$^{-1}$) (**f**, **g**), or HKLM (10$^7$ CFU ml$^{-1}$) (**h**, **i**) for different times; Cypj abundance and the phosphorylation of the indicated proteins (**d**, **f**, **h**), as well as transcription of *Tnf* and *Il-6* (**e**, **g**, **i**; *N* = 3) were detected. Error bars indicate S.D.; * *p* < 0.05, ** *p* < 0.01 (two-tailed Student's *t*-test)

**C-terminal region of CYPJ interacts with TAB2 and TAB3**. In order to identify candidate target(s) in the NF-κB signal pathway that is/are regulated by CYPJ, a series of dual-luciferase reporter assays were performed by co-transfection of the NF-κB reporter and proteins of the NF-κB pathway into 293T cells with or without enforced CYPJ expression (Fig. 3a). CYPJ significantly inhibited the NF-κB reporter activation induced by TRAF6, TAK1-TAB1, TAK1-TAB2, but not NIK, p65-p50, or IKKα (SS/EE), a constitutively active mutant of IKKα (Fig. 3a). These results suggest that CYPJ

blocks classic NF-κB signaling by acting downstream of TRAF6 and upstream of the IKK complex, and the TAK1-TAB complex is a potential target. The activation of TAK1-TAB protein complex stimulates a series of signal pathways including NF-κB and AP-1[25,29]. We then performed luciferase assays and revealed that CYPJ inhibited both the TRAF6 and TAK1-TAB1/2-activated AP-1 reporter (Supplementary Fig. 6).

We carried out co-immunoprecipitation (co-IP) assays to screen for binding partners of CYPJ in NF-κB signal pathway,

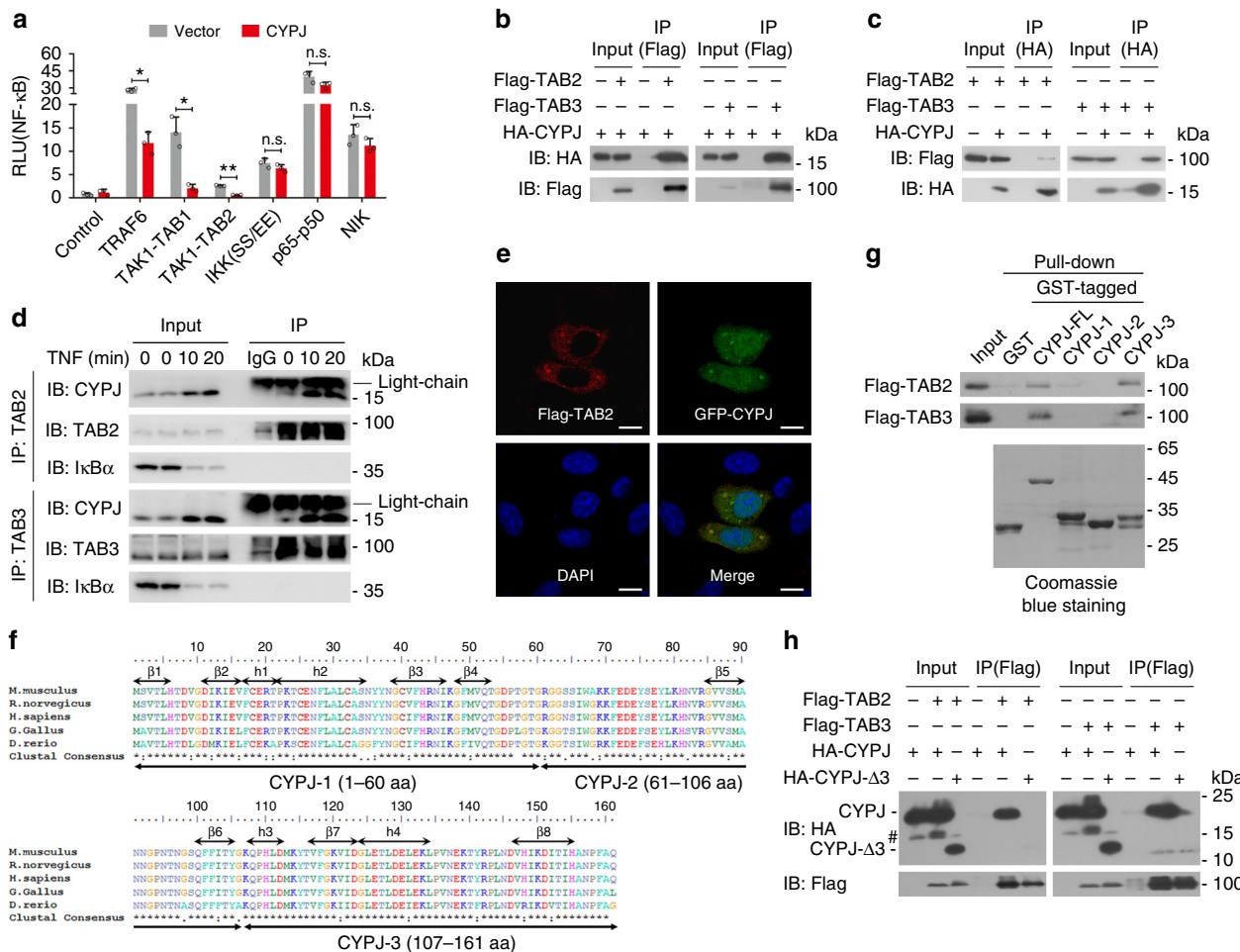

**Fig. 3** The C-terminal of CYPJ interacts with TAB2 and TAB3. **a** Effects of CYPJ on NF-κB reporter activation induced by individually transfected components of the NF-κB signal pathway ($N = 3$). **b, c** Reciprocal co-IP assays determine the interaction between HA-CYPJ and Flag-TAB2 or Flag-TAB3 in 293T cells. **d** Endogenous co-IP assays evaluate the interaction between CYPJ and TAB2/TAB3 with or without TNF treatment for different time points in 293T cells. **e** Immunofluorescence microscopy observing the colocalization of pEGFP-CYPJ and Flag-TAB2 in HeLa cells. Scale bars, 20 μm. **f** Multiple sequence alignments of CYPJ proteins from different species. The three fragments of CYPJ are marked. **g** GST pull-down assays were performed using *E. coli*-expressed GST and GST-CYPJ fragments and lysates of 293T cells transfected with Flag-TAB2 or Flag-TAB3. **h** Co-IP assays detecting the interaction between Flag-TAB2/3 with HA-tagged FL or fragment-3 deleted CYPJ, # indicates a nonspecific band. Error bars indicate S.D.; n.s. no significance, * $p < 0.05$, ** $p < 0.01$ (two-tailed Student's *t*-test)

including TAK1 and its associated and upstream and downstream molecules. Notably, we found that CYPJ specifically binds to TAB2 and TAB3 (Fig. 3b, c) but not TAK1, TRAF2, TRAF6, p50, or p65 (Supplementary Fig. 7a–e) in reciprocal co-IP assays. We performed endogenous co-IP to confirm the interactions of these proteins. Lysates of 293T cells, which were treated with or without TNF for different times, were immunoprecipitated with an anti-TAB2 antibody, an anti-TAB3 antibody or control IgG. Immunoblotting (IB) with an anti-CYPJ antibody confirmed the interaction between endogenous CYPJ and TAB2/TAB3 (Fig. 3d). For comparison, we also detected the binding of CYPA to TAB2/3 by reciprocal co-IP, and the results showed that CYPA was not associated with these two proteins in cells (Supplementary Fig. 7f), suggesting that CYPA suppresses NF-κB signal pathway via distinct mechanism. Although CYPJ is mainly localized in the nucleus, confocal microscopy also identified a fraction of the cytoplasm distributed GFP-CYPJ that co-localized with Flag-TAB2 (Fig. 3e). Furthermore, competition co-IP showed that CYPJ overexpression does not influence the interactions between TAK1-TAB2 (Supplementary Fig. 8a) and TAK1-TAB3 (Supplementary Fig. 8b).

To narrow down the region of CYPJ that mediates its binding to TAB2 and TAB3, we divided CYPJ into three segments based on its structure[32] (Fig. 3f). GST-fused full-length (FL) or fragmented CYPJ (CYPJ-1/2/3) and GST proteins were expressed and purified from *Escherichia coli* strain BL21. These prokaryotically expressed proteins were incubated with lysates of Flag-TAB2/TAB3-transfected 293T cells in GST pull-down assays, and the results showed that fragment-3 of CYPJ (CYPJ-3) strongly interacted with both TAB2 and TAB3 (Fig. 3g). We generated a truncated CYPJ with a deletion of fragment-3 (CYPJ-Δ3) to evaluate whether this region is required for CYPJ to interact with TAB2/3. Co-IP experiments showed that CYPJ, but not CYPJ-Δ3, binds to TAB2 and TAB3 (Fig. 3h). The above data demonstrated that the C-terminal region is indispensable for CYPJ to form a protein complex with TAB2 and TAB3.

**CYPJ binds to the NZF domains of TAB2 and TAB3.** Reciprocally, we co-transfected plasmids encoding full-length (FL) or truncated TAB2 (ΔCUE, ΔNZF, or ΔCC&NZF; Fig. 4a), along with HA-CYPJ into 293T cells, and the cell lysates were used for

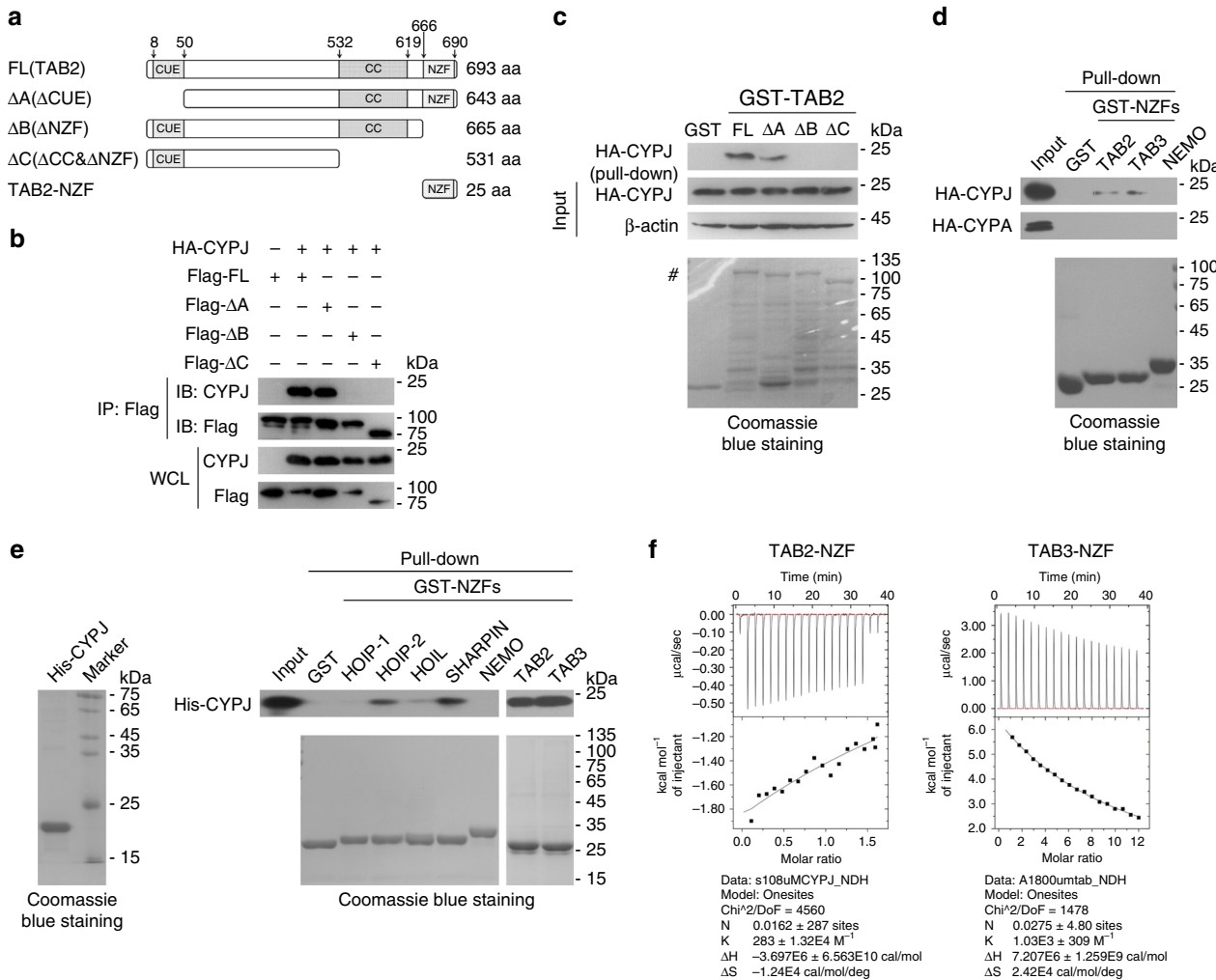

**Fig. 4** CYPJ binds to the NZF domain of TAB2 and TAB3. **a** Schematic diagram of FL and truncation variants of TAB2. **b** HA-CYPJ and Flag-tagged TAB2 fragments were co-transfected into 293T cells for co-IP assays to detect their interaction. **c** GST pull-down assays were performed using *E.coli*-expressed GST and GST-TAB2 truncations and lysates of 293T cells transfected with HA-CYPJ. # indicates the bands of GST-fused full-length and truncated TAB2. **d** *E.coli*-expressed GST and GST-fused NZF domains of TAB2, TAB3, and NEMO were used in GST pull-down assays with lysates of 293T cells transfected with HA-CYPJ or HA-CYPA. **e** *E.coli*-expressed GST and GST-fused NZF domains of LUBAC members, as well as TAB2 and TAB3 were used in GST pull-down assays with *E.coli*-expressed His-CYPJ. **f** ITC assays were performed to detect the binding of in vitro synthesized NZF peptides derived from TAB2 or TAB3 to prokaryotically expressed His-CYPJ

co-IP assays. The results showed that CYPJ bound to the FL and ΔCUE, but not the ΔNZF or ΔCC&NZF form of TAB2 (Fig. 4b). We expressed and purified GST-fused FL or truncated TAB2 (ΔCUE, ΔNZF, or ΔCC&NZF) from *E.coli* strain BL21 and incubated them with lysates of HA-CYPJ-transfected 293T cells in GST pull-down assays. A similar phenomenon was observed, in that deletion of the NZF domain completely blocked the interaction between CYPJ and TAB2 (Fig. 4c). These results suggest that the NZF domain of TAB2 is required for TAB2 to interact with CYPJ. We used prokaryotically expressed GST-fused NZF domains of TAB2, TAB3, and NEMO in GST pull-down assays with lysates of HA-CYPJ-transfected 293T cells, and the results indicated that CYPJ interacted with the NZF domains of TAB2 and TAB3 but not NEMO (Fig. 4d). In accordance with previous observation that CYPA did not form complex with TAB2 and TAB3 (Supplementary Fig. 7f), we confirmed that HA-CYPA does not bind to *E.coli* expressed GST-NZF domain of TAB2 or TAB3 (Fig. 4d). As components of the LUBAC also contain NZF domains[35,36], we further detected their binding specificity for CYPJ. Prokaryotically expressed recombinant His-

CYPJ was used in GST pull-down assays with the GST-fused NZF domains of HOIP, HOIL-1L, SHARPIN, NEMO, TAB2, and TAB3. The results showed that CYPJ strongly interacts with the NZF domains of TAB2, TAB3, HIOP (the second NZF domain), and SHARPIN (Fig. 4e). Based on this observation, we performed reciprocal co-IP and confirmed that CYPJ is associated with HOIP in 293T cells (Supplementary Fig. 9a). An isothermal titration calorimetry assay (ITC), in turn, found direct binding of in vitro synthesized TAB2-NZF and TAB3-NZF peptides to recombinant His-CYPJ (Fig. 4f). These results demonstrate the NZF domains of TAB2 and TAB3 are targets of CYPJ in mammalian cells.

**CYPJ disrupts the recruitment of the K63-Ub chain to TAB2/3.** The NZF domain of TAB2 and TAB3 is important for the sensing and recruitment of the TAK1-TAB complex to the K63-Ub chain during inflammation[21]. Modification of the TAB2/3 NZF domain by the bacterial methyltransferase NleE[26] or occupation of the TAB3 NZF domain by mycobacterium PtpA[27] disrupts the

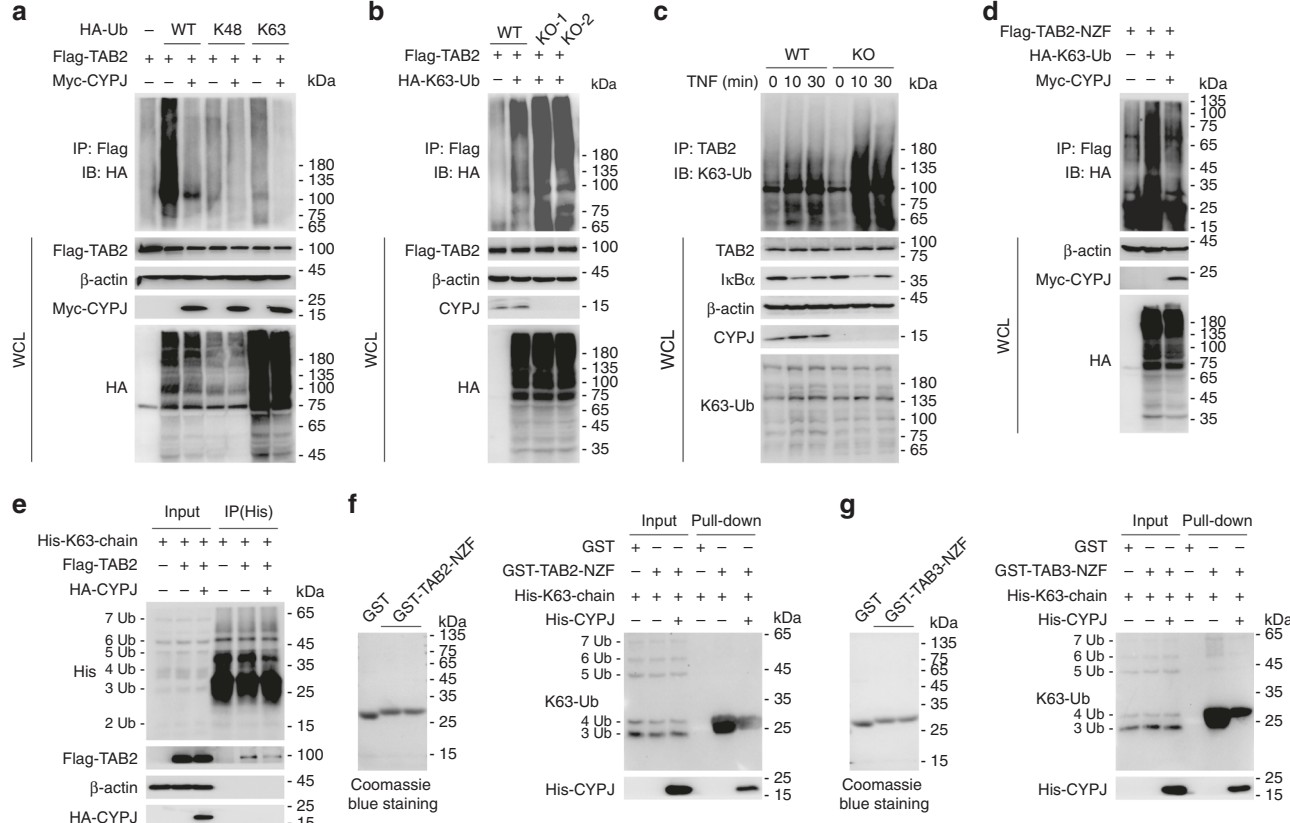

**Fig. 5** CYPJ disrupts the K63-linked ubiquitin-chain sensing of TAB2 and TAB3. **a** 293T cells were transfected with plasmids as indicated for 24 h. The enforced expressed Flag-TAB2 was immunoprecipitated with an anti-Flag antibody, and the polyubiquitin signals were detected with an anti-HA antibody. **b** WT or *CYPJ*-knockout (KO-1 and KO-2) 293T cells were transfected with or without HA-K63-Ub and Flag-TAB2 for 24 h, followed by IP and IB as indicated. **c** Lysates of WT or *CYPJ*-knockout 293T cells treated with TNF for different times were immunoprecipitated with an anti-TAB2 antibody, followed by IB with the indicated antibodies, including anti-K63 Ub. **d** 293T cells were co-transfected with Flag-TAB2-NZF, HA-K63-Ub, and Myc-CYPJ as indicated, and cell lysates were immunoprecipitated with an anti-Flag antibody followed by IB with an anti-HA antibody. **e** Lysates of Flag-TAB2 with or without HA-CYPJ-transfected 293T cells were incubated with in vitro synthesized His-K63-Ub chain (1 μg) as indicated. The His-K63-chain was immunoprecipitated with an anti-His antibody, and the bound Flag-TAB2 was detected with an anti-Flag antibody. **f**, **g** Binding of GST, GST-CYPJ, or GST-TAB2-NZF (**f**) or GST-TAB3-NZF (**g**) to the His-K63-chain was monitored by GST pull-down assays as indicated. Approximately 1 μg of the His-K63-chain was used in each reaction

binding of the K63-Ub chain and thus negatively regulates NF-κB signaling and host defense. We questioned whether CYPJ could regulate this important process. We overexpressed Flag-TAB2 and HA-Ub (encoding wide-type, K48-linked ubiquitin, or K63-linked ubiquitin, respectively) in 293T cells, with or without Myc-CYPJ co-transfection. After immunoprecipitation with an anti-Flag antibody and IB with an anti-HA antibody, we found that enforced CYPJ expression greatly reduced polyubiquitin signal in the K63-Ub-transfected group (Fig. 5a). Conversely, CYPJ deficiency (KO-1 and KO-2) increased the binding of K63-linked polyubiquitin chain to Flag-TAB2 (Fig. 5b). Next, we evaluated the endogenous polyubiquitin signal by immunoprecipitation with a TAB2 antibody followed by IB with a K63-Ub-specific antibody. As shown in Fig. 5c, TNF treatment increased the binding of K63-linked polyubiquitin chain to TAB2, which was largely enhanced by CYPJ deficiency in 293T cells. In another experiment, we co-transfected 293T cells with plasmids encoding Flag-TAB2-NZF and HA-K63-Ub, with or without enforced Myc-CYPJ expression. The results confirmed that CYPJ disrupted TAB2-NZF mediated K63-Ub chain sensing (Fig. 5d).

Subsequently, we applied in vitro synthesized His-tagged K63-Ub chain (His-K63-chain, 2–7Ub) to evaluate the function of CYPJ to disrupt the binding of the K63-Ub chain to TAB2/3. We incubated His-K63-chain with lysates of 293T cells transfected with Flag-TAB2 with or without enforced CYPJ expression (Fig. 5e). The His-K63-chain was immunoprecipitated with an anti-His antibody, and associated Flag-TAB2 was detected with an anti-Flag antibody. The results showed that CYPJ significantly blocked the binding of TAB2 to the His-K63-Ub chain (Fig. 5e). To exclude the possibility that CYPJ forms a complex with the polyubiquitin chain and thus disrupts its binding to TAB2, we expressed and purified GST-fused CYPJ and TAB2-NZF and used the purified proteins in GST pull-down assays with His-K63-chain. As shown in Supplementary Fig. 10, the His-K63-chain specifically interacts with TAB2-NZF but not CYPJ. We mixed the recombinant His-K63-Ub chain and GST-TAB2/3-NZF, with or without bacterial expressed and purified His-CYPJ, and GST pull-down assays indicated that CYPJ significantly reduced the recruitment of the K63-Ub chain to TAB2-NZF (Fig. 5f) or TAB3-NZF (Fig. 5g). On the other hand, the NZF domains of the LUBAC components are important for the synthesis of linear ubiquitin chain which is conjugated to NEMO in the activation of the canonical NF-κB pathway[37]. As CYPJ forms a complex with HOIP (Supplementary Fig. 9a) and the NZF domain of several components of the LUBAC (Fig. 4e), we overexpressed CYPJ in 293T cells and confirmed its inhibitory effect on the linear ubiquitination of NEMO (Supplementary Fig. 9b), which is a well-known LUBAC substrate. At the mean time, enforced CYPJ

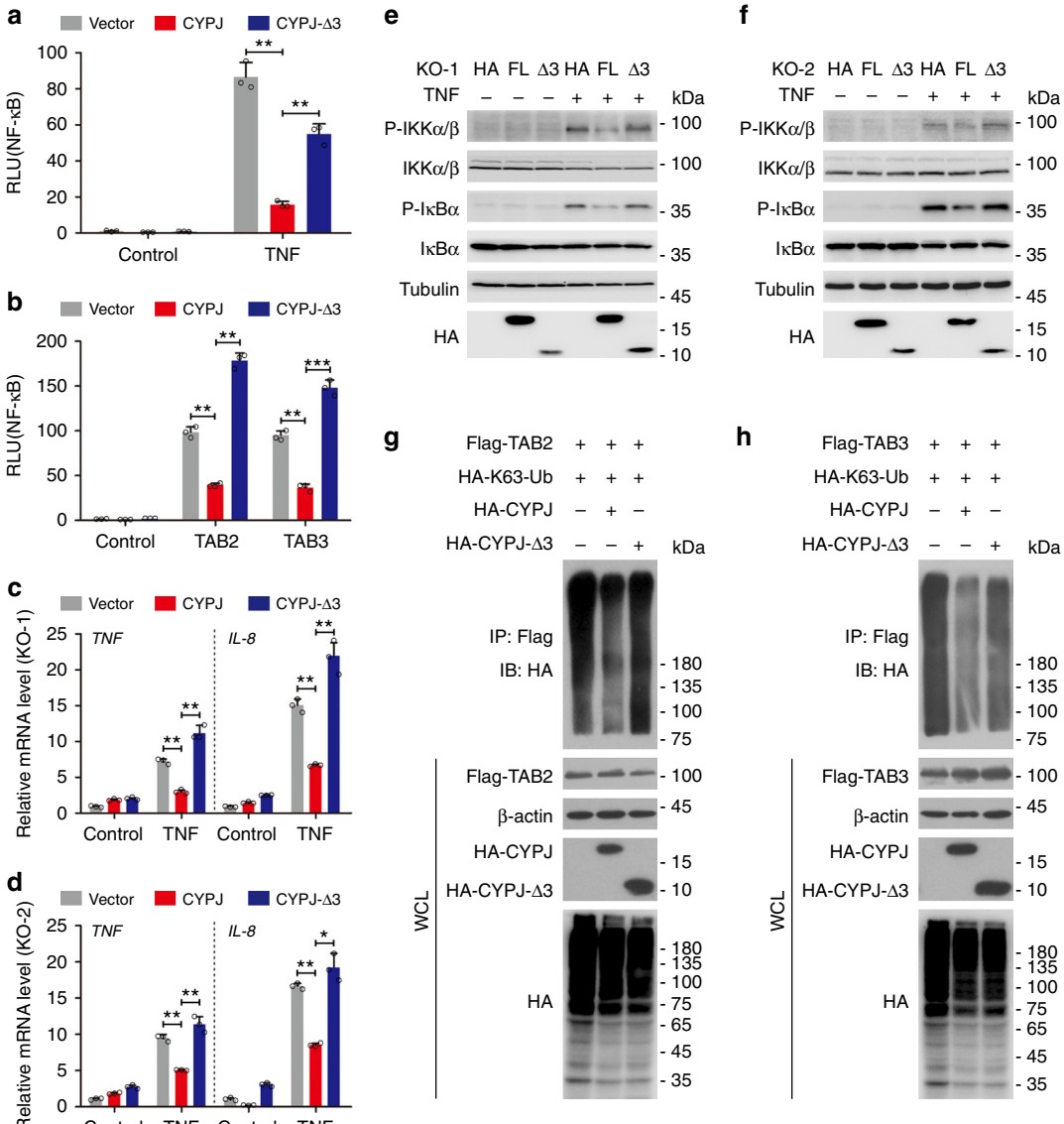

**Fig. 6** The C-terminus is required for CYPJ to inhibit NF-κB signaling. **a, b** Effects of CYPJ and CYPJ-Δ3 on the activity of NF-κB reporters induced by TNF (**a** $N = 3$) or TAB2/3 (**b** $N = 3$). **c, d** Effects of CYPJ and CYPJ-Δ3 on TNF-induced transcription of *TNF* and *IL-8* in the CYPJ-knockout 293T cell lines KO-1 (**c** $N = 3$) and KO-2 (**d** $N = 3$). **e, f** Effects of CYPJ and CYPJ-Δ3 overexpression on TNF-induced phosphorylation of indicated proteins in two CYPJ-knockout cell lines cell lines, KO-1 (**e**) and KO-2 (**f**). **g, h** Fragment-3 is required for CYPJ to disrupt the binding of K63-Ub chain to TAB2 (**g**) and TAB3 (**h**). Plasmids were transfected into 293T cells as indicated and the binding of polyubiquitin chain to TAB2/3 was detected with an anti-HA antibody after IP with an anti-Flag antibody. Error bars indicate S.D.; * $p < 0.05$, ** $p < 0.01$, *** $p < 0.001$ (two-tailed Student's *t*-test)

expression attenuated HOIP-induced activation of the NF-κB reporter (Supplementary Fig. 9c). These results indicate that CYPJ also functions as a negative regulator of LUBAC.

**The C-terminus of CYPJ is indispensable for its inhibition**. Given that the C-terminus of CYPJ mediates its binding TAB2 and TAB3 (Fig. 3g, h), we questioned whether this region is required for the inhibitory function of CYPJ. We overexpressed CYPJ and CYPJ-Δ3 in 293T cells and found that this deletion greatly reduced the ability of CYPJ to inhibit NF-κB activation induced by TNF (Fig. 6a) or TAB2/3 overexpression (Fig. 6b). Next, we overexpressed CYPJ and CYPJ-Δ3 in CYPJ-knockout 293T cell lines (KO-1 and KO-2). The full-length CYPJ, but not the fragment-3-deleted truncation, significantly repressed TNF-induced *TNF* and *IL-8* transcription in both cell lines (Fig. 6c, d).

Fragment-3 deletion also attenuated the ability of CYPJ to suppress TNF-induced IKKα/β and IκBα phosphorylation (Fig. 6e, f). Subsequently, we monitored the ability of fragment-3 of CYPJ to block K63-Ub chain sensing. Immunoprecipitation assays showed that full-length CYPJ blocked the binding of K63-Ub chain to TAB2 (Fig. 6g) and TAB3 (Fig. 6h), and this function of CYPJ is largely reduced by the deletion of fragment-3 (Fig. 6g, h). These findings demonstrated that the C-terminal sequence is critically required for CYPJ to compete with the association between K63-Ub chain and TAB2/3, which mediates the activation of the NF-κB signal pathway.

**Cypj protects mice from spesis and DSS-induced colitis**. Finally, we assessed the biological significance of Cypj in controlling inflammatory responses in vivo using Cypj-deficient

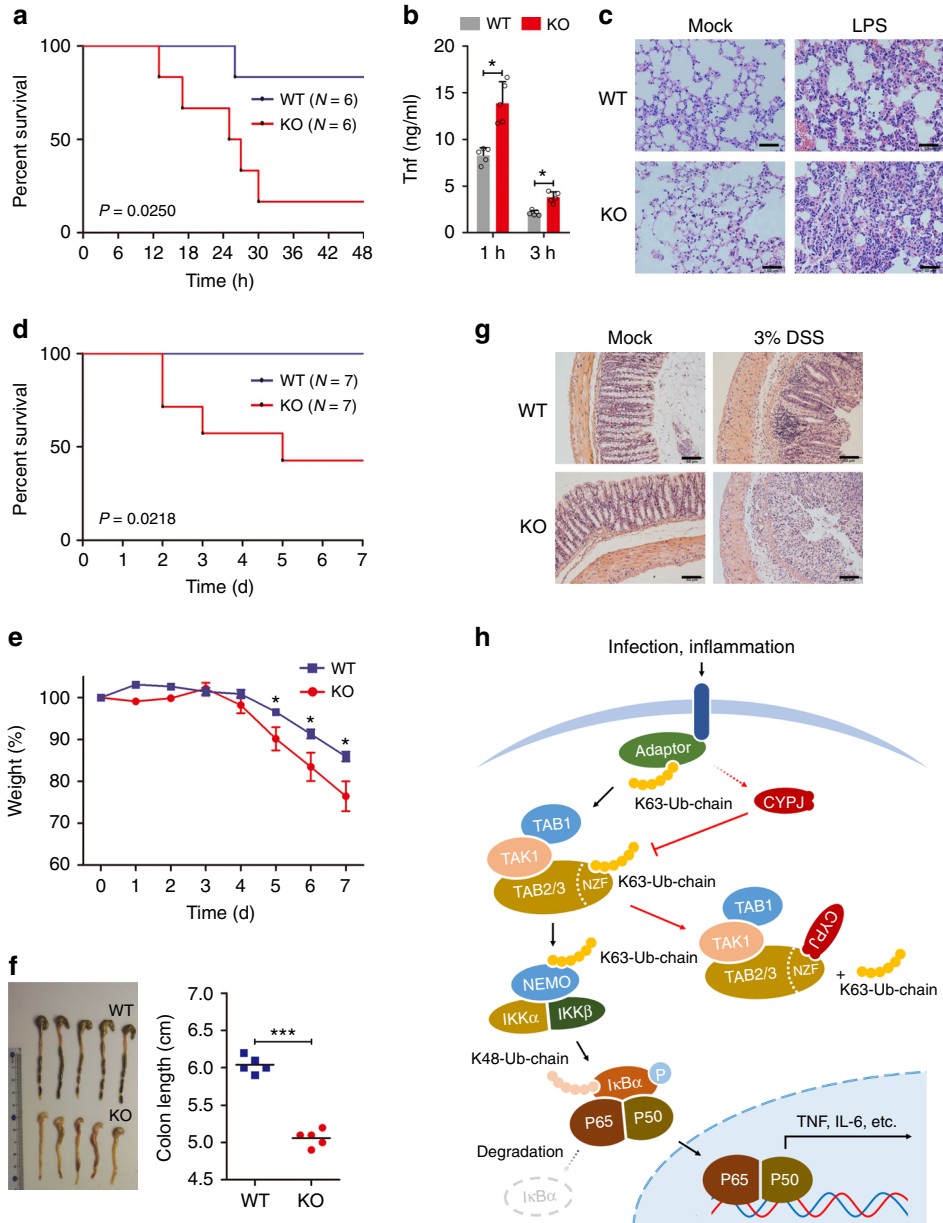

**Fig. 7** Cypj protects mice from lethal acute inflammation and DSS-colitis. **a**, **b** Survival curve (**a** $N = 6$ mice per group, Log-rank Test) and ELISA of serum Tnf level (**b** $N = 5$ mice per group, two-tailed Student's *t*-test) in Cypj-deficient and WT mice after intraperitoneal injection of LPS (10 mg per kg body weight). **c** Histopathology of lungs from Cypj-deficient and control mice after intraperitoneal injection of LPS (10 mg kg$^{-1}$). Scale bars, 100 μm. **d** Survival curve ($N = 7$ mice per group, Log-rank test) of WT and Cypj KO mice after intraperitoneal injection of HKLM ($2 \times 10^{12}$ CFU per mice). **e** Weight loss during 3% DSS-induced colitis was expressed as the percentage of change from day 0 ($N = 5$ mice per group, two-tailed Student's *t*-test). **f** Colon lengths of DSS-treated mice ($N = 5$ mice per group, two-tailed Student's *t*-test) were assessed at the time of necropsy, with representative macroscopic images and statistics. **g** Representative micrographs of colon hematoxylin and eosin (H&E) staining from WT and KO mice 7 days after DSS-colitis. Scale bars, 100 μm. **h** Proposed working model of CYPJ dependent negative feedback regulation of the NF-κB signal pathway. Error bars indicate S.D.; * $p < 0.05$, *** $p < 0.001$

mice. Wild-type (WT) and knockout (KO) mice were injected intraperitoneally with a sub-lethal dose of LPS (10 mg kg$^{-1}$), and survival time and the production of proinflammatory cytokines were compared between these two groups. Five out of six Cypj-deficient mice died within 48 h (83.3%) with a median survival time of ~24 h, while there is only one mouse died in the WT group (16.7%; Fig. 7a). Moreover, Cypj-deficient mice produced higher levels of Tnf (Fig. 7b), and their lungs exhibited more severe tissue damage and diffuse inflammation (Fig. 7c). In other experiments, intraperitoneal injection of LPS at 0 h (5 mg kg$^{-1}$)

and 72 h (50 mg kg$^{-1}$) sequentially (Supplementary Fig. 11a), or intraperitoneal injection of HKLM ($2 \times 10^{12}$ CFU per mouse, Fig. 7d) both showed similar results, in that the survival time of Cypj KO mice was significantly shorter than that of their WT counterparts.

To further investigate the physiological role of Cypj in inflammation regulation, we applied a mouse colitis model using DSS. Although KO mice did not exhibit aberrant colonic morphology and inflammation before the treatment (Supplementary Fig. 11b), they were more susceptible to 3% DSS-induced

colitis and displayed exaggerated weight loss compared to their WT counterparts (Fig. 7e). The KO mice presented a significant increase in colonic ulceration and occult bleeding, whereas the WT mice displayed only mild inflammation and no gross ulceration (Fig. 7f). Additionally, the colon length of KO mice was shorter than that of WT mice (Fig. 7f). Histological analyses revealed that Cypj deficiency resulted in a more severe disruption of mucosal structures of the colon, as compared with control mice (Fig. 7g).

Taken together, the above results indicate an important role of Cypj in inflammation resolution in vivo.

## Discussion

Ubiquitin, a small highly conserved protein composed of 76 amino acid residues, plays a critical role in NF-κB signaling. It contains 7 lysines (K6, K11, K27, K29, K33, K48, and K63) and forms a polyubiquitin chain through covalent bonds between one ubiquitin and the K of another ubiquitin[13]. Ubiquitination depends on the covalent ligation of ubiquitin to a lysine residue in the target protein and is catalyzed by three enzymes: ubiquitin-activating enzyme (E1), ubiquitin-conjugating enzyme (E2), and ubiquitin-ligasing enzyme (E3)[5]. Unlike the complex biological process of ubiquitination, there exists another simple combination mode that relies on ubiquitin-binding domains (UBDs), including the NZF domain in TAB2 and TAB3, which is independent of lysine. Binding of TAB2 or TAB3 to K63-linked polyubiquitin chains via their NZF domain recruits the TAK1-TABs kinase complex to the polyubiquitin chain and activates downstream signal pathways[19–21]. We identify CYPJ, a member of the PPIase family, as a novel regulator of the NF-κB signal pathway. Co-IP and GST pull-down assays demonstrated that CYPJ binds to the NZF domain of TAB2 and TAB3. The C-terminal region of CYPJ mediates this binding, and truncated CYPJ with a deletion of fragment-3 completely abolished the interaction between CYPJ and TAB2/TAB3. Further experiments showed that CYPJ disrupts the interaction between K63-Ub and the NZF domain, and this effect was largely abolished by the deletion of fragment-3. These findings demonstrate a critical role of the C-terminal region of CYPJ in regulating NF-κB signaling.

Negative feedback regulation of the NF-κB signal pathway is an important mechanism to maintain homeostasis of innate immunity[8]. Several regulators have been reported to regulate this process by manipulating the ubiquitin system, and their defects are associated with human diseases. For example, the deubiquitinase A20 and CYLD are induced in response to cytokine treatment and infection and have been found to inhibit NF-κB by hydrolyzing K63-linked polyubiquitin chains on key NF-κB signaling molecules, such as RIP1, TRAF6, and TAK1[38]. Defects of A20 and CYLD are found in several types of cancer, which leads to the hyperactivation of NF-κB and triggers uncontrolled growth and metastasis of tumor cells[39]. Elevated expression of CYPJ protein in response to inflammatory stimulation, with its mRNA unchanged, notably inhibits the activation of the NF-κB reporter, expression of proinflammatory cytokines, phosphorylation of IKKα/β and, nuclear translocation of p65. In contrast, Cypj knockout increased Iκbα phosphorylation and Tnf and Il-6 mRNA abundance in response to LPS, Tnf, HKLM, and Il-1β in mouse BMMs. Our findings reveal a model of CYPJ-mediated negative feedback regulation of the NF-κB signal pathway by disrupting K63-Ub chain sensing of TAB2/3 (Fig. 7h) and mitigating LUBAC activity (Supplementary Fig. 12).

It has been reported that PPIases are associated with viral infection and innate immunity. CYPA is associated with the Gag protein of HIV-1 and is incorporated into viral particles[40]. CYPB interacts with NS5B of hepatitis C (HCV) and facilitates its replication[41]. In contrast, the interaction between CYPA and the influenza M1 protein restricts viral replication[42]. Inhibitors of cyclophilin, especially nonimmunorepressive analogs of cyclosporine A, have been reported to have antiviral activity[29,43]. Several studies also reported the regulatory effects of cyclophilin on innate immune responses. For example, CYPA accelerates IFN-β production via the regulation of RIG-I ubiquitination and inhibits IAV replication[44]. CYPA can also be secreted, and extracellular CYPA was considered to function as a cytokine that promotes tumorigenesis through its receptor CD147[45]. Interestingly, our results show that both CYPJ and CYPA inhibit NF-κB signaling independent of their PPIase activity. Cypj knockout protects mice from lethal endotoxin shock, HKLM treatment, and DSS-induced colitis. The diverse and complicated relationships between CYPs and innate immunity require further study.

Taken together, our findings provide a new mechanism of CYPJ-mediated negative feedback regulation of the NF-κB signal pathway that maintains homeostasis of innate immune responses and shed light on the development of novel therapeutic approaches to control inflammation.

## Methods

**Mice**. The Cypj-deficient mice were generated by Shanghai Biomodel Organism Science & Technology Development Co.,Ltd (Shanghai, China) using CRISPR/Cas9 gene targeting technology. The genotyping of Cypj-knockout mice was confirmed by sequencing of PCR fragments (373 bp) from genomic DNA isolated from 2–3 mm tail tips using the following primers: Forward 5′-GAGCTGCTTTATGT-GAACAAGCC-3′ and Reverse 5′-AGGACAGGTCTCACTCTGTAGCTCT-3′. LPS from E.coli, 011:B4 (L2630) was purchased from Sigma-Aldrich. Murine M-CSF (AF-315-02), murine TNF (AF-315-01), and murine IL-1β (AF-211-11B) were ordered from Reprotech. LPS was used at a final concentration of 100 ng ml$^{-1}$ for cells treatment while 10 or 5/50 mg$^{-1}$ kg$^{-1}$ for animal experiments. The concentration of TNF and Il-6 in serum was measured by ELISA Kit from R&D Systems. Dextran sulfate sodium (DSS, 36–50 kDa) (160110) was bought from MP Biomeicals. DSS-induced colitis was induced in WT or KO C57BL/6 mice with 3% (wt/vol) DSS dissolved in drinking water. Mice were monitored for body weight every day. H&E staining was performed by Google Biotechnology (Wuhan, China). All animal experiments were undertaken in line with the National Institute of Health Guide for the Care and Use of Laboratory Animals and maintained under SPF (specific-pathogen-free) condition. The protocols have been approved by the Animal welfare and Ethics Committee of Sun Yat-sen University Cancer Center and Guangdong Pharmaceutical University. All mice used were 6 to 8 weeks of age. The experiments were randomized and the investigators were blinded to allocation.

**Plasmids and virus**. Plasmids encoding RIG-I, MAVS (VISA/ISP-1), TRAF2, TRAF6, TAK1, TAB1, TAB2, NIK, NF-κB-luc, and AP-1-luc were provided by Prof. Xin Ye at Institute of Microbiology, Chinese Academy of Sciences (Beijing, China). The IKKα expression plasmid was shared by Prof. Hongbin Shu at Wuhan University (Wuhan, China). Expression vectors for HA-tagged wild-type Ub, K48/K63-Ub, and GST-TAB2/3-NZF plasmids were from Prof. Feng Shao at National Institute of Biological Sciences (Beijing, China). The TAB2 truncations ΔCUE, ΔNZF, ΔCC&ΔNZF, NZF were generated by subcloning the corresponding sequences into pcDNA3.0/Flag plasmid (Invitrogen). CYPJ and CYPA expression plasmids were generated by subcloning into pCMV-HA/pCMV-Myc vector. The CYPJ truncations CYPJ-1, CYPJ-2, and CYPJ-3 were constructed into pGEX-4T-1 vector (GE Healthcare) and GFP-CYPJ-Δ1, GFP-CYPJ-Δ2, and GFP-CYPJ-Δ3 were constructed into pEGFP-N1 vector (CLONTECH Laboratories). GST-fused NZF domains of LUBAC complex were synthesized by Synbio Technologies Inc (Suzhou, China) and then subcloned cDNA sequences into the pGEX4T-1 plasmid (GE Healthcare). CYPJ-mut (R44A&F49A) and CYPA-mut (R55A&F60A) were also synthesized by Synbio Techologies Inc (Suzhou, China). The influenza A virus (WSN) was propagated in the allantoic fluid of 10-day-old embryonied chicken eggs. GFP-VSV was amplified in Vero cells. For in vivo assay, listeria monocytogenes carrying ovalbumin were gifted by Prof. Penghui Zhou at Sun Yat-sen University Cancer Center (Guangzhou, China) and heated-inactivated by 70 °C water bath for 30 min.

**Antibodies and reagents**. The rabbit antibody CYPJ/PPIL3 (1:1000; ab169936), Ubiquitin (linkage-specific K63) (1:1000; ab179434), and TAB3 (1:1000; ab85655) was purchased from Abcam. The rabbit polyclonal antibody PPIL3 (1:1000; C-term, AP17078b) were bought from ABGENT. The his6-polyUb WT Chains (2–7, K63-linked) (1:1000; UCH-330) was purchased from Boston Biochem, R&D systems. The mouse anti-Flag (1:2000; F3165) antibody was ordered from Sigma. The antibodies against CYPA (1:2000; #2175), P-IKKα/β (1:800; Ser176/180, #2697), P-IκBα (1:1000; Ser 32/36, #9246), P-p38 (1:1000; Thr180/Tyr182, #4511), Erk1/2

(1:2000; #46955), P-Erk1/2 (1:2000; Thr202/Tyr204,#4370), P-JNK (1:1000; Thr183/Tyr185,#4668), JNK (1:1000; #9252), and human tumor necrosis factor-α (#8902) was purchased from Cell Signaling Technology. The cell lysis buffer (10×) (#8903) and RIPA buffer (10×) (#8906) were also purchased from Cell Signaling Technology. The mouse antibodies specific for c-Myc (9E10) (1:1000; SC-40), HA-probe (H-7) (1:1000; SC-7392), β-actin (C4) (1:1000; sc-47778), Ub (P4D1) (1:1000; SC-8017), and rabbit antibody IKKα/β (H-470) (1:1000; SC-7607) and goat antibody TAB2 (K-20) (1:1000; SC-11851) were from Santa Cruz Biotechnology. The mouse antibodies GAPDH (1:2000; RM2002V), β-Tubulin (1:2000; RM2003V), His (1:1000; RM1001V), Myc (1:1000; RM1003V), and HA (1:1000; RM1004V) were bought from New Progress Biotechnology (Guangzhou, China). The rabbit Myc (1:1000; A00172) and goat HA (1:1000; A00168) antibodies were bought from Genscript (Nanjing, China). The mouse antibodies specific for NF-κB-P65 (F-6) (1:1000; ZS-8008) and p38 (1:1000; ZS-71490) were purchased from Zhongshan Golden Bridge Biotechnology (Beijing, China). Anti-mouse GFP (1:2000; 66002-1-Ig), anti-rabbit GFP (1:1000; 50430-2-AP), β-actin (1:2000; 66009-1-Ig), TAB2 (1:1000; 14410-1-AP), and PCNA (1:2000; 60097-1-Ig) were from Proteintech. RELA/p65 (1:100; A10609) monoclonal antibody for immunofluorescence was got from Abclonal. Secondary antibodies were purchased from Jackson (1:5000; Jackson ImmunoResearch Inc.). Glutathione Sepharose 4B (17-0756-01) and Protein G beads (SC-2002) were ordered from GE Healthcare and Santa Cruz Biotechnology, respectively. DAPI staining kit (KGA215) was bought from KeyGEN Biotechnology (Nanjing, China). Protein Marker (M221) was bought from Genstar. Dual-Luciferase Reporter Assay System was ordered from Promega. The isolation kit of cytoplasm and nucleus (A00172) was purchased from Beyotime (Shanghai, China). 30% acrylamide-bisacrylamide (FD2065) and 5 × Dual loading buffer (FD006) were ordered from FDbio science (Hangzhou, China). The Super ECL detection reagent was purchased from KeyGEN Biotechnology (Nanjing, China). CD3e (1:100; 45-0031-82), CD8a (1:100; 25-0081-82), CD19 (1:100; 17-0193-82), and NKP46 (1:100; 12-3351-82) were purchased from eBioscience. NK1.1 (1:100; 561111), CD11c (1:100; 560584), and MHCII (1:100; 562352) were purchased from BD. F4/80 (1:100; 123116) and CD11b (1:100; 101208) were bought from Biolegend Biosciences. HKLM (tlrl-hklm) was ordered from InvivoGen.

**Cell culture and transfection**. 293T cells (human embryonic kidney cell line), HeLa cells (human epithelial carcinoma cell line), and Vero cells were cultured in Dulbecco's modified Eagle's medium (DMEM) (Gibco) supplemented with 10% fetal bovine serum (Gibco, 10270-106) at 37 °C under 5% CO₂. A549 cells (human type II alveolar epithelial cell line) was cultured in Roswell Park Memorial Institute 1640 (Gibco) supplemented with 10% fetal bovine serum (Gibco, 10270-106). 293T, HeLa, Vero, and A549 were originally obtained from ATCC and all human cell lines were authenticated by China Center for Type Culture Collection (CCTCC). Bone marrow derived macrophages (BMM) were separated from tibia and femur of C57BL/6 mice and cultured in conditional medium (1640 + 10% FBS + 1% GlutaMAX™ + 100 Uml⁻¹ penicillin + 100 μg ml⁻¹ streptomycin + 20 ng ml⁻¹ M-CSF) for 5–7 days and prepared following experiments. No cell lines used in this study were found in the database of commonly misidentified cell lines that is maintained by ICLAC and NCBI Biosample. Mycoplasma contamination was routinely checked by PCR analysis and eliminated by Plasmocin™ treatment (ant-mpt). The primers for mycoplasma detection: Myco-F: 5′-GGGAGCAAACAGGA TTAGATACCCT-3′; Myco-R: 5′-TGCACCATCTGTCACTCTGTTAACCTC-3′. Indicated plasmids were transfected into cells with DNA Transfection Reagent B35101 (Bimake) or Lipofectamine™ 2000 (Invitrogen) according to the manufacturer's instructions.

**Dual-luciferase reporter assay**. Cells were transfected with plasmids encoding NF-κB or AP-1 luciferase reporter together with pRL-TK and appropriate plasmids. Cells were collected and lysed 24 h later. Subsequently, luciferase activity was measured with the Dual-Luciferase reporter assay system (Promega) according to the manufacturer's protocols. Data were normalized by the ratio of firefly luciferase activity to renilla luciferase activity. Each set of assays was performed in triplicate.

**RNAi**. The sequences of human CYPJ double-stranded small interfering RNAs (siRNAs) were designed as follows: si-CYPJ-1 target sequence 5′-CTGGAAGAG GAGGCAACAG-3′; si-CYPJ-2 target sequence 5′-CCGACCTCTTAATGATGTA-3′. The sequences for mice Cypa/j siRNAs were as follows: si-Cypj-1 target sequence 5′-GGATCTCAGTTCTTCATCA-3′; si-Cypj-2 target sequence 5′-AGAT AGAAGTCTTCTGTGA-3′; si-Cypa-1 target sequence 5′-CCATCTACGGAGAG AAATT-3′; si-Cypa-2 target sequence 5′-GCATCTTGTCCATGGCAAA-3′. All the siRNA oligonucleotides containing 3′dTdT overhanging sequences were chemically synthesized in RIBOBIO (Guangzhou, China) and transfected into cells using Lipofectamine™ 2000 (Invitrogen) or Lipofectamine™ RNAiMAX Transfection Reagent (thermo fisher scientific) according to the manufacturer's instructions. A negative control nucleotide (si-control) was also purchased from RIBOBIO (Guangzhou, China).

**293T CYPJ-KO and CYPA-KO cells**. KO-CYPJ/CYPA 293T cells were based on CRISPR/Cas9 system according to ZHANG Feng's lab. The sequences used in

sgRNA targeting specific genes (http://crispr.mit.edu) were as follows: sg-CYPJ-1F 5′-CACCGGAGGACACCCAAAACATGTG-3′; sg-CYPJ-1R 5′-AAACCACATGTT TTGGGTGTCCTC-3′; sg-CYPJ-2F 5′-CACCGTGACACTGCATACAGATGT-3′; sg-CYPJ-2R 5′-AAACACATCTGTATGCAGTGTCAC-3′; sg-CYPA-1F 5′-CACC GACTGCCAAGACTGAGTGGTA-3′; sg-CYPA-1R 5′-AAACTACCACTCAGTC TTGGCAGTC-3′; sg-CYPA-2F 5′-CACCGTCACCCACCCTGTCAACATAT-3′; and sg-CYPA-2R 5′-AAACATATGTTGACAGGGTGGTGAC-3′. The sgRNA sequences were constructed into lentiCRISPR V2 (Addgene, 52961) and positive pool cells were screened by 2 μg ml⁻¹ puromycin for 2–4 days. Monoclone was screened by infinite diluting the pool cells and seed 30–50 cells into one 96-plate. The KO-monoclone was following identified by sequencing and IB analysis. The primers for genome DNA (TIANGEN, DP304) PCR were as follows: CYPJ-genome-PCR-F 5′-CCACACAGGGAAATAAACAATG-3′; CYPJ-genome-PCR-R 5′-GGATGGCTAGTTTTCCCTAACA-3′; CYPA-genome-PCR-F 5′-ATTTTAT TTGGGGTTGCTCCCTT-3′; and CYPA-genome-PCT-R 5′-GCCATGTTG-TACCCTTACCACT-3′.

**qRT-PCR**. Total RNA was extracted with TRIzol (Invitrogen) according to procedural guidelines. cDNA was synthesized using the EasySript One-step gDNA Removal and cDNA Synthesis kit (TRANSGEN AE311, Beijing, China). Universal iQ™ SYBR Green Super mix (Bio-Rad) was used and performed in CFX96 (Bio-Rad). The cycle conditions included an initial denaturation step at 95 °C for 3 min followed by 40 cycles of amplification for 5 s at 95 °C, 30 s at 60 °C. A melting curve and standard curve were run to evaluate the amplification specificity and efficiency, respectively. The cycle threshold values were used to calculate the relative mRNA expression levels. The date was normalized to the housekeeping gene GAPDH in each individual sample and $2^{-\Delta\Delta Ct}$ was used to calculate relative expression changes. Specific primers used for qRT-PCR assays are shown in Supplementary Table 1.

**Flow cytometry**. Mature BMM and BMDC were harvested by cell scraper and washed with PBS for three times, followed by staining with indicated antibodies for 30 min on ice. Fresh mouse spleen was gently ground into single-cell suspension and filtered by cell strainer (40 μm, BD Falcon). Total cells were washed with cold PBS for three times followed by staining with indicated antibodies for 30 min on ice. After washed three times and resuspended in 1% FBS in PBS buffer, cells were collected on BD LSRFortessa™ X-20 Cell Analyzer and analyzed by BD FACSDiva™ Software (BD, USA). All flow cytometry gating strategies are shown in Supplementary Fig. 13.

**Immunoprecipitation and immunoblot**. Cells were prepared by washing with cold PBS and then lysed with 1× lysis buffer (Cell Signaling Technology) for 1 h. The supernatants were collected after centrifugation 12,000 rpm for 10 min at 4 °C, and then incubated with appropriate antibodies for 1–2 h followed by adding 20–30 μl protein G beads (Santa Cruz Biotechnology) at 4 °C overnight. The beads were washed three times with IP buffer (50 mM Tris-HCl (pH 7.4), 150 mM NaCl and 1% Nonidet P40), followed by immunoblot (IB) analysis. For the IB assay, cells were harvested and lysed with 1 × RIPA buffer (Cell Signaling Technology), followed by samples were boiled for 5 min together with 2 × or 5 ×loading buffer to be used to perform SDS-PAGE. Proteins were further transferred onto 0.22 μm PVDF membrane (Roche). The membrane was sealed with 5% fat-free milk in Tris-buffered saline added 1% Tween-20 (TBS-T) for 1–2 h at room temperature and then incubated with appropriate primary antibody at 4 °C overnight. The membrane was washed three times for 10 min each with TBS-T and then incubated with a horseradish peroxidase (HRP)-conjugated secondary antibody for 1 h at room temperature. After washing three times with TBS-T, the membrane was flushed with super ECL (KeyGEN, Nanjing, China) or Immobilon Western HRP (Millipore). Then the bands were exposed by X-Ray film (FUJI, JAPAN) or detected by ChemiDoc Touch (Bio-Rad). Uncropped scans are shown in Supplementary Fig. 14.

**Ubiquitin assay**. Cells were transfected with plasmids encoding HA-Ub or its mutants, Flag-TAB2/3 or mutants with or without Myc-CYPJ, followed by collecting and lysing cells using 1 × cell lysis buffer (Cell Signaling Technology) adding protease inhibitor cocktail (Bimake). Then the cell supernatant was incubated with 1 μg Flag antibody (Sigma) and following 20–30 μl protein G beads at 4 °C overnight. Beads were washed three times with IP buffer. The samples were boiled for 5 min together with 2× loading buffer and then performed SDS-PAGE, followed by transferred onto PVDF membrane, sealed with 5% milk and incubated with primary and secondary antibodies.

**GST pull-down assay**. GST and GST fusion proteins were expressed and purified from E. coli BL21 strain (DE3) using Glutathione Sepharose 4B (GE healthcare). Associated plasmids were transfected into 293T cells for 24 h, and then collected and lysed the cells by using 1× lysis buffer (Cell Signaling Technology). The lysates were incubated with GST or GST-fused protein at 4 °C overnight. The beads were washed three times with PBS or IP buffer and further boiled for 5 min with 2× loading buffer. Prepared samples were analyzed by IB with appropriate antibody.

**Immunofluorescence staining for confocal microscopy**. For the CYPJ and TAB2 colocalization analysis, plasmids encoding GFP-CYPJ and Flag-TAB2 were co-transfected into HeLa cells for 24 h. Mouse anti-Flag antibody was used as a primary antibody and Alexa Fluor 594-conjugated anti-mouse IgG (Zhongshan Golden Bridge Biotechnology) as a secondary antibody. The nucleus was stained with DAPI (KeyGEN Biotechnology). For the P65 nuclear localization analysis, HeLa cells were transfected with GFP-CYPJ for 24 h and then stimulated with TNF (15 ng ml$^{-1}$, Cell Signaling Technology) for 15 min or unstimulated as control. The nucleus was stained with DAPI (KeyGEN Biotechnology). The cells were then washed three times with PBS, fixed in 4% paraformaldehyde for 20–30 min at room temperature and permeabilized with 0.2% Triton X-100 for 5 min. After blocking in 5% BSA for 30 min, the cells were incubated for 2 h with an anti-p65 monoclonal antibody at 4 °C. After washing with PBS for three times, the cells were incubated for 45 min with Alexa Flour 594-conjugated anti-mouse IgG antibody. The final result was observed by using laser confocal fluorescence microscopy (ZEISS LSM880, Germany).

**Polyubiquitin chain assay**. For eukaryotic polyubiquitin chain binding assay, the 293T cells were co-transfected with FLAG3.0/FLAG-TAB2 and HA/HA-CYPJ. After 24 h, the cells were harvested and lysed for another 1 h. Then the supernatants were incubated with 1 μg of K63 Ub chain and equivalent anti-His for 2 h, followed by adding 20–30 μl protein G beads (Santa Cruz Biotechnology) at 4 °C overnight. The beads were washed three times with IP buffer, followed by IB analysis.

For prokaryotic polyubiquitin chain pull-down assay, 1 μg of K63 Ub chain, 5 μg His-CYPJ, and moderate GST or GST-TAB2/3-NZF were incubated at 4 °C overnight. The beads were washed three times with IP buffer and analyzed by IB using K63-Ub or His antibody.

**ITC assay**. TAB2-NZF (QWNCTACTFLNHPALIRCEQCEMP) and TAB3-NZF (WNCDSCTFLNHPALNRCEQCEMP) were synthesized in DgPeptides co., ltd (Hangzhou, China). His-CYPJ was expressed form *E.coli* BL21 strain (DE3) and purified by Ni-NTA His Bind Resin (PAN001-001C) (7sea Biotech, Shanghai, China). ITC assay was performed on MicroCalTM iTC200 (GE Healthcare) according to standard procedure.

**Statistical and reproducibility**. All experiments were performed at least three times. Statistical analyses were performed using the GraphPad Prism Software and two-tailed Student's *t*-test with a *P* value < 0.05 was used to identify the significance of these data. Survival curve was performed via the Log-rank (Mantel-Cox) test and *p* value of <0.05 was considered statistically significant.

## Data availability

The authors declare that the data supporting the findings of this study are available within the article and its supplementary information files, or are available upon reasonable requests to the authors. Raw data that support the findings of this study have also been deposited in Research Data Deposit public platform (http://www.researchdata.org.cn), with the Approval Number as RDDB2018000413.

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

## Acknowledgements

This work was supported in part by grants from the National Natural Science Foundation of China (81672744), the Sci-Tech Project Foundation of Guangzhou City (2018030100003 and 201707020039), the Sci-Tech Project Foundation of Guangdong Province (2016A020217007) and the National Basic Research Program of China (973 Program, 2012CB518903).

## Author Contributions

C.S. performed most of the experiments and analysis. C.Y. and Z.W. performed mice experiments and generated mutation constructs. H.C. helped with mice maintain and animal facilities. D.X., Y.Z., and H.H. provided technical assistance. S.C. and W.H. conceived the study. S.C. and C.S. wrote the manuscript.

## Additional information

**Competing interests:** The authors declare no competing interests.

