## [Peer Review File · Nature Communications]

Reviewers' comments:

Reviewer #1 (TAK1/TAB signaling)(Remarks to the Author):

The authors report that cyclophilin J is a novel negative regulator of NF- κ B through blocking TAB2/TAB3 binding to poly-ubiquitin chain. While the finding may potentially reveal a new regulatory mean in NF- κ B signaling, the data shown do not yet sufficiently support the authors' conclusions. At this point, the study is preliminary.

Major concerns

1) The in vivo data shown in Fig. 7 are not convincing. Differences between wild type and CYPJ-deficient mice in DSS colitis model and LPS sepsis model are marginal, which is far less pronounced compared with those by deficiency of the well established other negative regulators of NF- κ B, e.g. A20 and CYLD. This raises a major concern whether CYPJ plays any physiological roles in inflammation. The authors should demonstrate convincing effect of Cypj deficiency, if any, by using other conditions or inflammation models.

2) Supplementary Figure S1C demonstrates that inhibitory activity against NF- κ B is not specific to CYPJ. CYPJ is equally effective, which raises a number of questions; e.g. whether both redundantly inhibit NF- κ B; whether CYPJ also binds to NZF and whether other cyclophilins share this activity. The authors should include results of CYPJ in other figures, and examine whether Cypj deficiency enhances NF- κ B signaling.

3) While overexpressed CYPJ is clearly capable of binding to TAB2/TAB3, the results in Fig. 3D are not compelling to demonstrate the interaction between TAB2/TAB3 and CYPJ at endogenous protein levels. Reciprocal immunoprecipitation and a time course analysis should be provided.

4) Ubiquitin binding data are all used overexpressed ubiquitin, which is not sufficient. A time course analysis of how TNF-induced TAB2/3 binding to poly-ubiquitin chain is changed in wild type and Cypj-deficient cells should be examined without exogenous proteins.

5) The authors should also test whether CYPJ does not alter interaction between TAB2/3 and TAK1.

Other concerns

i) Fig.4 and 5 should include the input protein levels of ubiquitin (poly ubiquitin chain).

ii) Fig. 2D-- CYPJ immunoblotting should be included. The upregulation of IKK activation by Cypj deficiency is marginal, which does not support the conclusion.

iii) The title is misleading. The results do not show that CYPJ disrupts poly-ubiquitin chain

iv) Cecum seems to be very small in Cypj deficient mice, which is unusual and needs a discussion.

ii) Editing is required.

Reviewer #2 (Inflammation signaling and animal models)(Remarks to the Author):

Manuscript 'Cyclophilin J disrupts lysine 63-linked ubiquitin chain sensing in inflammation resolution', by Sheng et al. identifies CYPJ as a negative feedback regulator of NF- κ B signaling by binding to TAB-2 and -3 disrupting the binding of K63-Ub chains to these proteins.

- The study is informative and data are overall convincing. However, the manuscript is poorly written with lots of grammatical and linguistic mistakes.

- The study completely neglects the importance of linear (M1) ubiquitination for NF- κ B signalling. CYPJ maybe interferes with K63-Ub signaling, however, M1 signaling at least needs to be discussed here.
- Fig. 1g : nuclear p65 staining unclear to me.
- Fig. 1i : CYPJ expression is induced after stimulation of Hela, BMM and A549 cells : is CYPJ an NF- κ B response gene ?
- Fig. 2d, f, h : the authors need to confirm absence of CYPJ expression in CRISPR knockout cells (Western blots). Page 12 "CRISPR-Cas9 edition results in a truncated form of CYPJ" : this is nowhere shown. Did the authors generate a knockout of a truncated CYPJ ?
- Fig. 5d : not convincing. Hard to believe the statement that CYPJ blocks the binding of TAB2 to K63-Ub based on these data.
- Fig. 6 : sublethal dose of LPS (30 mg/kg) : what is a sublethal dose when also 60 % of WT mice die. To me a sublethal dose is a dose to which WT mice all survive. I propose to lower the dose of LPS. Also: how do mice respond to a sublethal dose of TNF ?

Minor comments:

- TNF α should be replaced by TNF (since TNF β has been renamed to LT).
- Page 7, lines 139 and 144 : treatment of cells with IL-1 β (and not IL-6).

Reviewer #1:

1. The *in vivo* data shown in Fig. 7 are not convincing. Differences between wild type and CYPJ-deficient mice in DSS colitis model and LPS sepsis model are marginal, which is far less pronounced compared with those by deficiency of the well established other negative regulators of NF- κ B, e.g. A20 and CYLD. This raises a major concern whether CYPJ plays any physiological roles in inflammation. The authors should demonstrate convincing effect of Cypj deficiency, if any, by using other conditions or inflammation models.

Response: We thank the reviewer's concern. The unapparent differences between wild type and Cypj-deficient mice are largely due to the improper condition used in previous assay. We optimized the dose of these inflammatory reagents and presented better results in this revised manuscript.

In sub-lethal dose of 10 mg/kg LPS experiment, only 1 mouse died in WT group (1/6) within 48 h compared to 5 mice died in KO group (5/6; **Fig.7a**). In another experiment, we intraperitoneally injected LPS at 0 h (5 mg/kg) and 72 h (50 mg/kg) sequentially as shown in **Supplementary Fig. 11a**, and the data showed that all WT mice remained alive while 4 out of 7 KO mice died. In both endotoxin shock assays, the survival times between WT and KO mice are statistically significant, indicating Cypj plays important roles *in vivo*.

Compared to well established negative regulators, although the weight loss between WT and Cypj-KO mice looks less significant in DSS model (**Fig. 7e**), morphology and histological analyses showed that Cypj deficiency resulted in severer atrophy and disruption of mucosal structures of the colons (**Fig. 7f, g**). Moreover, the function of Cypj is comparable to that of newly identified inflammation resolver Tet2 (Zhang Q, et al. Tet 2 is required to resolve inflammation by recruiting Hdac2 to specifically repress IL-6. 2015 *Nature* **525**, 389-393, figure 1g). In addition, we used HKLM (heat killed *Listeria Monocytogenes*) to induce systematic inflammation. Intraperitoneal injection of HKLM (2×10^{12} CFU/mouse) showed that WT mice were all alive after 7 days while 4 out of 7 KO mice died (**Fig.7d**). Taken these *in vivo* data together, we concluded that Cypj is a new negative regulator of inflammation.

2. *Supplementary Figure SIC demonstrates that inhibitory activity against NF- κ B is not specific to CYPJ. CYPJ is equally effective, which raises a number of questions; e.g. whether both redundantly inhibit NF- κ B; whether CYPJ also binds to NZF and whether*

other cyclophilins share this activity. The authors should include results of CYPA in other figures, and examine whether Cypa deficiency enhances NF-κB signaling.

Response: We agree with the reviewer that it's important to compare the activity of CYPJ with other cyclophilins, such as CYPA.

In the revised manuscript, we provided evidences to show that the protein abundance of both CYPJ and CYPA are induced by inflammatory stimulations such as TNF- α (**Fig. 1i**), LPS (**Fig. 1j**) and VSV infection (**Fig. 1k**). Luciferase assay showed that CYPJ and CYPA, as well as their catalytic dead mutants, attenuate TNF- α induced NF- κ B activation (**Supplementary Fig. 1c**). We deleted *CYPA* in genome of 293T cells by CRISPR-Cas9 technique (**Supplementary Fig. 1f**, inset), and found that CYPA deficiency significantly enhanced NF- κ B activation (**Supplementary Fig. 1f**), *TNF- α* and *IL-8* transcription (**Supplementary Fig. 1g**) and IKK α/β and I κ B α phosphorylation (**Supplementary Fig. 1i**) induced by TNF- α . Furthermore, we silenced *Cypa* or *Cypj* expression in mBMM cells by siRNA (**Supplementary Fig. 4a, 4b**). Compared to si-NC group, both *Cypa* and *Cypj* knockdown promoted secretion of proinflammatory cytokines *Tnf- α* and *Il-6* (**Supplementary Fig. 4c, 4d**). Based on these observations, we conclude that CYPJ and CYPA are negative regulators of NF- κ B signaling pathway.

In this manuscript we provided a mechanism that CYPJ binds to NZF domain of TAB2 and TAB3 thus disrupts the recruitment of K63-Ub chain and inhibits NF- κ B signaling. As the reviewer questioned whether CYPA shared similar mechanism, we performed GST pull-down assay to address this issue. As shown in **Fig. 4c**, 293T expressed HA-CYPA does not bind to bacterially expressed and purified GST-NZF domain of TAB2, TAB3 or NEMO. At the mean time, CYPA does not interact with TAB2 or TAB3 in co-IP assays (**Supplementary Fig. 7f**). These results suggest that the mechanism of CYPA is distinct from that of CYPJ in regulating NF- κ B.

3. While overexpressed CYPJ is clearly capable of binding to TAB2/TAB3, the results in Fig. 3D are not compelling to demonstrate the interaction between TAB2/TAB3 and CYPJ at endogenous protein levels. Reciprocal immunoprecipitation and a time course analysis should be provided.

Response: We appreciate the reviewer's concern. We performed endogenous co-IP to confirm the interaction between CYPJ and TAB2/3 (**Fig. 3d**). 293T cells were treated with TNF- α for different times, and cell lysates were carried out for immunoprecipitation (IP) with anti-TAB2 or anti-TAB3 antibody. The amount of CYPJ forms complex with TAB2/3 was detected by WB using an anti-CYPJ antibody. The results confirmed endogenous interaction between CYPJ and TAB2/3, and the binding between these molecules increased with TNF- α treatment in a time dependent manner (**Fig. 3d**). Unfortunately we cannot find an anti-CYPJ antibody suitable for immunoprecipitation, so it is regrettable that the reciprocal co-IP could not be carried out.

4. Ubiquitin binding data are all used overexpressed ubiquitin, which is not sufficient. A time course analysis of how TNF-induced TAB2/3 binding to poly-ubiquitin chain is

changed in wild type and Cypj-deficient cells should be examined without exogenous proteins.

Response: We thank the reviewer's constructive concern. We generated CYPJ-deficient 293T cells by CRISPR-Cas9, and treated WT and KO cells with TNF- α for 0, 10 or 20 min. Then endogenous TAB2 was immunoprecipitated by an anti-TAB2 antibody and its associated K63-Ub chains were detected by a K63-Ub specific antibody. As shown in **Fig. 5c**, CYPJ deficiency increased TNF- α induced endogenous K63-Ub signal of TAB2.

5. The authors should also test whether CYPJ does not alter interaction between TAB2/3 and TAK1.

Response: We appreciate the reviewer's suggestion. We performed competitive co-IP in 293T cells transfected with plasmids as indicated in **Supplementary Fig. 8**. The results showed that all proteins were successfully expressed and CYPJ overexpression does not affect the interaction between TAK1-TAB2 (**Supplementary Fig. 8a**) and TAK1-TAB3 (**Supplementary Fig. 8b**).

6. Other concerns

- i) Fig. 4 and 5 should include the input protein levels of ubiquitin (poly ubiquitin chain).*
- ii) Fig. 2D-- CYPJ immunoblotting should be included. The upregulation of IKK activation by Cypj deficiency is marginal, which does not support the conclusion.*
- iii) The title is misleading. The results do not show that CYPJ disrupts poly-ubiquitin chain*
- iv) Cecum seems to be very small in Cypj deficient mice, which is unusual and needs a discussion.*
- ii) Editing is required.*

Response: We thank the reviewer's helpful concerns.

- i) The figures need to include input ubiquitin levels are mainly in **Fig. 5 and 6**. We found out the input samples from -80°C freezer, and performed WB using anti-HA to detect overexpressed Ub (**Fig. 5a, b, g and Fig. 6g, h**). Besides, we also detected the endogenous ubiquitin using an antibody that specifically recognizes K63-Ub chain (**Fig. 5c**).*
- ii) We purchased a new antibody to detect the endogenous Cypj (ABGENT AP17078b, C-term) and using samples of **Fig. 2d, f, h** and **Supplementary Fig. 3c** stored in -80°C freezer. The levels of P-Ikk α / β and Ikk α / β were detected at the mean time by WB. The results showed that the Cypj expression is absent in KO mBMMs, and P-Ikk α / β is significantly higher in KO cells in response to LPS (**Fig 2d**), Tnf- α (**Fig 2f**), HKLM (**Fig 2h**) or Il-1 β (**Supplementary Fig. 3c**). These results support our conclusion that Cypj is a negative regulator of NF- κ B signaling.*
- iii) In this manuscript, we provided evidences to show that CYPJ disrupts the interaction between K63-Ub-chain and NZF domains of TAB2/3 (**Fig. 5**). In these experiments, we also included synthesized K63-Ub-chain in several experiments to do *in vitro* binding assay with *E.coli*-expressed and purified CYPJ and NZF (**Fig. 5e,f**). The results showed that CYPJ reduces the binding of K63-Ub-chain to TAB2/3-NZF in a*

cell-free system (**Fig. 5e,f**). As these results strongly support our conclusion, we did not change the title of our manuscript.

- iv) The size of cecum between WT and KO mice without DSS treatment is similar (**Supplementary Fig. 11b**). The possible reason for the size of cecum looks smaller in KO group in **Fig. 7f** is that DSS induced stronger intestinal inflammation and severer diarrhea in KO mice, which resulted in less fecal storage in cecum.
- v) We carefully revised the manuscript. The quality of the language has been largely improved.

Reviewer #2:

1. The study is informative and data are overall convincing. However, the manuscript is poorly written with lots of grammatical and linguistic mistakes.

Response: We carefully revised the manuscript and largely improved the quality of the language.

2. The study completely neglects the importance of linear (M1) ubiquitination for NF- κ B signaling. CYPJ maybe interferes with K63-Ub signaling, however, M1 signaling at least needs to be discussed here.

Response: We appreciate the reviewer's suggestion. We modified **Fig.4d** and added new experiments in **Supplementary Fig.9** to evaluate the regulation of M1 ubiquitination.

We expressed and purified GST-fused NZF of HOIP-1, HOIP-2, HOIL, SHARPIN and NEMO from *E.coli*, and subsequently incubated these proteins with recombinant His-CYPJ in GST pull-down assay (**Fig.4d**). His-CYPJ binds to HOIP-2 and SHARPIN with higher affinity (**Fig.4d**), and the interaction between CYPJ and HOIP was confirmed by reciprocal co-IP experiments (**Supplementary Fig. 9a**). These results suggest that CYPJ forms complex with LUBAC (linear ubiquitin chain assembly complex) in cells.

Next, we investigated the regulation of CYPJ on LUBAC. HA-CYPJ overexpression in 293T cells largely reduced linear ubiquitination of NEMO, which is a major substrate of LUBAC (**Supplementary Fig. 9b**). As predicted, CYPJ enforced expression attenuates HOIP induced NF- κ B activation (**Supplementary Fig. 9c**). All these experiments indicated that CYPJ is a negative regulator of M1 ubiquitination.

3. Fig. 1g : nuclear p65 staining unclear to me.

Response: We performed the experiment again and replaced **Fig.1g** for better resolution.

4. Fig. 1i : CYPJ expression is induced after stimulation of Hela, BMM and A549 cells : is CYPJ an NF- κ B response gene ?

Response: We appreciate the reviewer's concern. We found out samples of **Fig. 1i-1k** from -80°C freezer and evaluated the mRNA abundance of *CYPJ* and its paralogous gene *CYPA* in response to inflammatory stimulations for different time (**Supplementary Fig. 2**). Although under these conditions the transcription of NF-κB downstream targets *TNF-α*, *IL-8* or *Il-6* were significantly induced, we are interested to find that the mRNA levels of *CYPJ* and *CYPA* kept unchanged (**Supplementary Fig. 2**). However, the protein of *CYPJ* and *CYPA* are induced (**Fig. 1i-1k**). These results suggest that *CYPJ* and *CYPA* are inflammation response proteins but the regulation is at post-transcription level. In other words, we cannot categorize *CYPJ* and *CYPA* to targets of NF-κB. The mechanism for inflammation induced expression of *CYPJ* and *CYPA* protein needs further study.

5. *Fig. 2d, f, h : the authors need to confirm absence of CYPJ expression in CRISPR knockout cells (Western blots). Page 12 "CRISPR-Cas9 edition results in a truncated form of CYPJ" : this is nowhere shown. Did the authors generate a knockout of a truncated CYPJ ?*

Response: We appreciate the reviewer's suggestion. From the schematic diagram of **Fig. 2a**, CRISPR-Cas9 genomic edition results in an in-frame stop codon at the 18th amino acids of *Cypj*. We found out the previous input samples and detected *Cypj* expression using a new *CYPJ* antibody (ABGENT AP17078b, C-term). As shown in **Fig. 2d, f, h** and **Supplementary Fig. 3a**, the full-length of *Cypj* expression is detected in WT mBMMs but is absent in KO cells. So we deleted the sentence "CRISPR-Cas9 edition results in a truncated form of *CYPJ*" from the revised manuscript which may mislead readers.

6. *Fig. 5d : not convincing. Hard to believe the statement that CYPJ blocks the binding of TAB2 to K63-Ub based on these data.*

Response: In **Fig. 5d**, we observed that less Flag-TAB2 binds to His-K63-chain in the presence of HA-CYPJ enforced expressed in 293T cells. In order to provide more evidences to support this conclusion, we added synthesized K63-Ub-chain to do *in vitro* binding assay with *E.coli*-expressed and purified His-CYPJ and GST-NZF. The results of **Fig. 5e and 5f** show that the amount of TAB2/3-NZF bound His-K63-chain is largely decreased in the present of *CYPJ* in a cell-free system. These results confirmed that *CYPJ* blocks the interaction between K63-Ub-chain and NZF domain of TAB2/3.

7. *Fig. 6 : sublethal dose of LPS (30 mg/kg) : what is a sublethal dose when also 60 % of WT mice die. To me a sublethal dose is a dose to which WT mice all survive. I propose to lower the dose of LPS. Also: how do mice respond to a sublethal dose of TNF ?*

Response: We appreciate the reviewer's concern. We decreased the dose of LPS and performed several new assays. In sublethal dose of 10 mg/kg LPS only 1 mouse died in WT group (1/6) compared to 5 mice died in KO group (5/6) in 48 h, and the difference between these two groups is statistically significant (**Fig.7a**). In another experiment, we intraperitoneal inject 5 mg/kg LPS at day 0 followed with 50 mg/kg intraperitoneal injection of LPS at day 3 (**Supplementary Fig. 11a**). The data showed that all 7 WT mice remained alive while 4 out of 7 KO mice died within 144 h.

We appreciate the reviewer's suggestion to evaluate the response of mice to a sublethal dose of TNF. As it's inconvenient to get murine Tnf- α , we chose another inflammatory agent HKLM (heat killed *Listeria Monocytogenes*) to do the experiment by instead. At a dosage of 2×10^{12} CFU/mouse, intraperitoneal injection of HKLM killed 4 out of 7 KO mice within 7 days while the WT mice were all live (**Fig.7d**). These results supported our conclusion that *Cypj* protected mice from acute systematic inflammation.

8. *TNF α should be replaced by TNF (since TNF β has been renamed to LT).*

Response: We thank the reviewer's suggestion. Both TNF- α and TNF are used to describe this cytokine in scientific papers. After carefully searching the PubMed website, we found that TNF- α is used more widely in inflammation-related papers. So we did not make the change in the revised manuscript.

9. *Page 7, lines 139 and 144 : treatment of cells with Il-1 β (and not IL-6).*

Response: We apologize for the mistakes. They are corrected in this revised manuscript (Page 8, lines 1 and 5).

REVIEWERS' COMMENTS:

Reviewer #1 (Remarks to the Author):

The authors have added new results that addressed the major concerns raised in the previous version. However, I still have a couple of comments that would improve the manuscript.

I disagree with the authors rebuttal statement regarding the title. The results showing CYJ is involved in downregulation of NF-kB are very limited (Fig. 1i-j), which are not very strong data. There are no in-depth mechanistic studies to determine how CYJ is regulated. No data of CYJ upregulation in the in vivo setting was shown. Overall, this manuscript does not focus on the process "in inflammation resolution". It is more appropriate to say that CYPJ limits inflammation.

Some statements on poly-ubiquitin chains are very confusing. I believe that the authors wish to investigate that binding of NZF to the polyubiquitin chains but not covalent modification of NZF with ubiquitins ("ubiquitination"). The authors should carefully review and revise the statements that use a word "ubiquitination".

Some of the authors' statements are not well thought out. "Proper regulation of the innate immune response is critical to the host, yet the mechanism is poorly understood" may be too exaggerated or too much simplified. There are a number of defined regulatory mechanisms in inflammation.

Minor point: Figure S10 labeling IP: His should be pull-down GST.

Reviewer #2 (Remarks to the Author):

The authors have addressed most of my concerns.

Some remaining remarks:

- I insist in using TNF in stead of TNFalpha. I know that TNFalpha can still be found in many publications. However, the nomenclature has been changes years ago (decades even) in order to make things easier. TNFbeta no longer exists (since it has been renamed LT), hence TNFalpha should no longer be used !
- Since CYPJ not only restricts K63 linkages but also M1 chains, why not mentioning this in title and abstract ?

Reviewer #1:

1. I disagree with the authors rebuttal statement regarding the title. The results showing CYJ is involved in downregulation of NF-kB are very limited (Fig. 1i-j), which are not very strong data. There are no in-depth mechanistic studies to determine how CYJ is regulated. No data of CYJ upregulation in the in vivo setting was shown. Overall, this manuscript does not focus on the process “in inflammation resolution”. It is more appropriate to say that CYPJ limits inflammation.

Response: We thank the reviewer’s suggestion. The title of the manuscript has now been changed to “Cyclophilin J limits inflammation through the blockage of ubiquitin chain sensing”.

2. Some statements on poly-ubiquitin chains are very confusing. I believe that the authors wish to investigate that binding of NZF to the polyubiquitin chains but not covalent modification of NZF with ubiquitins (“ubiquitination”). The authors should carefully review and revise the statements that use a word “ubiquitination”.

Response: The experiments of poly-ubiquitin chain binding are enriched in Fig.5 and Fig.6. We carefully revised the text and avoided to use the term “ubiquitination” as the reviewer pointed out that the mechanism of this study is mainly focused in the binding of NZF to ubiquitin chains.

3. Some of the authors’ statements are not well thought out. “Proper regulation of the innate immune response is critical to the host, yet the mechanism is poorly understood” may be too exaggerated or too much simplified. There are a number of defined regulatory mechanisms in inflammation.

Response: We agree with the reviewer’s concern. The statement “yet the mechanism is poorly understood” has been removed in the revised manuscript (Page 3, line 4).

4. Minor point: Figure S10 labeling IP: His should be pull-down GST.

Response: We are sorry for the mistake. The labeling has been corrected in the revised Supplementary Figure 10.

Reviewer #2:

1. I insist in using TNF in stead of TNFalpha. I know that TNFalpha can still be found in many publications. However, the nomenclature has been changes years ago

(decades even) in order to make things easier. TNFbeta no longer exists (since it has been renamed LT), hence TNFalpha should no longer be used !

Response: We thank the reviewer's suggestion. The term TNF- α has been replaced by TNF in the revised manuscript.

2. Since CYPJ not only restricts K63 linkages but also M1 chains, why not mentioning this in title and abstract ?

Response: We thank the reviewer's suggestion. We added CYPJ interacts with "components of the linear ubiquitin chain assembly complex (LUBAC)" (Page 2, line 7) in the abstract. As NZF domain is required for the sensing of both K63 and M1 linked ubiquitin chain, we deleted the term "K63-linked" in the title to avoid the linkage restriction. The new title is "Cyclophilin J limits inflammation through the blockage of ubiquitin chain sensing".